# Regulation of the physiology and virulence of *Ralstonia solanacearum* by the second messenger 2′,3′-cyclic guanosine monophosphate

Xia Li[1,9], Wenfang Yin[2,9], Junjie Desmond Lin [3,9], Yong Zhang[4], Quan Guo[1], Gerun Wang[1], Xiayu Chen[1], Binbin Cui[1], Mingfang Wang [1], Min Chen[4], Peng Li[5], Ya-Wen He [6], Wei Qian [7], Haibin Luo [8], Lian-Hui Zhang[2], Xue-Wei Liu [3 ✉], Shihao Song [8 ✉] & Yinyue Deng [1 ✉]

Previous studies have demonstrated that bis-(3′,5′)-cyclic diguanosine monophosphate (bis-3′,5′-c-di-GMP) is a ubiquitous second messenger employed by bacteria. Here, we report that 2′,3′-cyclic guanosine monophosphate (2′,3′-cGMP) controls the important biological functions, quorum sensing (QS) signaling systems and virulence in *Ralstonia solanacearum* through the transcriptional regulator RSp0980. This signal specifically binds to RSp0980 with high affinity and thus abolishes the interaction between RSp0980 and the promoters of target genes. In-frame deletion of RSp0334, which contains an evolved GGDEF domain with a LLARLGGDQF motif required to catalyze 2′,3′-cGMP to (2′,5′)(3′,5′)-cyclic diguanosine monophosphate (2′,3′-c-di-GMP), altered the abovementioned important phenotypes through increasing the intracellular 2′,3′-cGMP levels. Furthermore, we found that 2′,3′-cGMP, its receptor and the evolved GGDEF domain with a LLARLGGDEF motif also exist in the human pathogen *Salmonella typhimurium*. Together, our work provides insights into the unusual function of the GGDEF domain of RSp0334 and the special regulatory mechanism of 2′,3′-cGMP signal in bacteria.

Nucleotide second messengers are a class of intracellular signaling molecules[1]. Among them, bis-(3′,5′)-cyclic diguanosine monophosphate (bis-3′,5′-c-di-GMP) has been found to be used as a second messenger in all major bacterial phyla[2]. It was first found in *Acetobacter xylinum*, in which it is involved in the regulation of cellulose synthesis[3]. Since then, bis-3′,5′-c-di-GMP has been shown to control a variety of cellular processes in bacteria[4–6]. This second messenger is synthesized by diguanylate cyclases (DGCs; characterized by GGDEF domains)

[1]School of Pharmaceutical Sciences (Shenzhen), Shenzhen Campus of Sun Yat-sen University, Sun Yat-sen University, Shenzhen, China. [2]Integrative Microbiology Research Center, College of Plant Protection, South China Agricultural University, Guangzhou, China. [3]Division of Chemistry and Biological Chemistry, School of Physical and Mathematical Sciences, Nanyang Technological University, Singapore, Singapore. [4]College of Resources and Environment, Southwest University, Chongqing, China. [5]Ministry of Education Key Laboratory for Ecology of Tropical Islands, Key Laboratory of Tropical Animal and Plant Ecology of Hainan Province, College of Life Sciences, Hainan Normal University, Haikou, China. [6]State Key Laboratory of Microbial Metabolism, Joint International Research Laboratory of Metabolic and Developmental Sciences, School of Life Sciences and Biotechnology, Shanghai Jiao Tong University, Shanghai, China. [7]State Key Laboratory of Plant Genomics, Institution of Microbiology, Chinese Academy of Sciences, Beijing, China. [8]Key Laboratory of Tropical Biological Resources of Ministry of Education, School of Pharmaceutical Sciences, Hainan University, Haikou, China. [9]These authors contributed equally: Xia Li, Wenfang Yin, Junjie Desmond Lin. ✉e-mail: xuewei@ntu.edu.sg; songsh@hainanu.edu.cn; dengyle@mail.sysu.edu.cn

from two GTPs and degraded by specific phosphodiesterases (PDEs; containing either EAL or HD-GYP domains)[7–9]. The bis-3′,5′-c-di-GMP signal transduction systems provide bacteria with the ability to sense changes in cell status or environmental conditions and execute appropriate physiological and social behaviors in response[10]. Various families of bis-3′,5′-c-di-GMP effectors have been identified and characterized to date, including PilZ domain proteins[4,11], degenerate GGDEF or EAL domain proteins[12,13], transcriptional regulators[14,15], mRNA riboswitches[16], and elongation factor P[17].

Quorum sensing (QS) signals are extracellular signaling molecules employed by many bacterial species to coordinate group behaviors in response to cell density[18,19]. To date, multiple QS signals have been identified in bacterial species, and recently, a common metabolic product, anthranilic acid, was revealed to be used as a bacterial signal[20,21]. Intriguingly, there is growing evidence that bacteria integrate QS and bis-3′,5′-c-di-GMP signaling to control biological functions. It was found that RpfR of *Burkholderia cenocepacia* is a QS signal receptor that also functions as a bis-3′,5′-c-di-GMP sensor[22,23], suggesting that bacterial pathogens use both intracellular and extracellular signals to integrate information about the physical and chemical surroundings and their population density to control their physiology and virulence characteristics.

In this study, we identified a protein, RSp0334, which contains a specific LLARLGGDQF motif that converts two 2′,3′-cyclic guanosine monophosphate (2′,3′-cGMP) molecules to one (2′,5′)(3′,5′)-cyclic diguanosine monophosphate (2′,3′-c-di-GMP) molecule. In-frame deletion of RSp0334 in *R. solanacearum* caused an increase in the intracellular level of 2′,3′-cGMP, which binds to its receptor RSp0980 and abolishes the interaction between RSp0980 and the promoters of target genes, ultimately impairing biofilm formation, motility, virulence and the production of QS signals. We also found that homologs of both RSp0334 and RSp0980 and intracellular 2′,3′-cGMP are present in the human pathogen *Salmonella typhimurium*, suggesting that the RSp0334/2′,3′-cGMP/RSp0980 signaling system is not solely existed in *R. solanacearum*.

## Results

### RSp0334 controls important biological functions in *R. solanacearum*

*R. solanacearum* possesses various GGDEF and EAL domain-containing proteins (Supplementary Data 1), which led us to investigate whether it employs bis-3′,5′-c-di-GMP or any other nucleotide second messenger. Our previous studies demonstrated that RpfR, which contains Per/Arnt/Sim (PAS)-GGDEF-EAL domains, is an important metabolic enzyme of bis-3′,5′-c-di-GMP that can sense both BDSF and bis-3′,5′-c-di-GMP[23]. To investigate the potential bis-3′,5′-c-di-GMP signaling system in *R. solanacearum*, we performed a homology search for RpfR from *Burkholderia cenocepacia* H111 in *R. solanacearum* GMI1000 by using the Basic Local Alignment Search Tool (BLAST) algorithm (https://blast.ncbi.nlm.nih.gov/Blast.cgi). We chose the top 9 homologs (in terms of identity), which share 37.79% to 42.63% identity with RpfR, for further study (Supplementary Data 2). In-frame deletion mutants of these 9 homologs were constructed. The deletion of *RSp0254*, *RSp0334* and *RSp1155* caused significant decreases in motility activity by 65.11%, 70.22% and 48%, respectively, compared to that in the *R. solanacearum* wild-type strain (Supplementary Fig. 1a). Moreover, biofilm formation was decreased by 50.42% in the *RSp0334* mutant strain but increased by 60.27% and 39.17% in the *RSp0254* and *RSp1155* mutant strains, respectively, compared to that in the wild-type strain (Supplementary Fig. 1b).

As deletion of *RSp0334* disrupted both biofilm formation and motility in *R. solanacearum*, it was selected for more detailed study of its role in regulating bacterial physiology (Fig. 1a, b). The deletion of *RSp0334* had little effect on the growth of bacterial cells in different media (Supplementary Fig. 1c–e) but resulted in significant defects in

phenotypes including motility, biofilm formation, cellulase production, and extracellular polysaccharide (EPS) production, and in trans expression of *RSp0334* restored all of the phenotypes of the *RSp0334* deletion mutant to those of the wild-type strain (Fig. 1c–f).

As the tested phenotypes are controlled by the anthranilic acid signaling and 3-hydroxypalmitic acid methyl ester (3-OH MAME) QS systems, we then tested whether there is a relationship between RSp0334 and these signaling systems. Intriguingly, the results of real-time quantitative reverse transcription PCR (RT–qPCR) and promoter-*lacZ* fusion reporter assays showed that the expression levels of *phcB*, *solI* and *trpEG* in the *RSp0334* mutant strain were significantly lower than those in the *R. solanacearum* GMI1000 wild-type strain (Fig. 1g–j). Consistent with these results, the levels of 3-OH MAME, N-octanoyl-L-homoserine lactone (C8-AHL), N-decanoyl-L-homoserine lactone (C10-AHL) and anthranilic acid production were reduced in the *RSp0334* mutant strain by 78.13%, 76.08%, 72.74% and 57.27%, respectively, and in trans expression of *RSp0334* almost completely rescued the production of these signals in the *RSp0334* mutant (Fig. 1k).

### RSp0334 negatively controls the intracellular 2′,3′-cGMP level in *R. solanacearum*

As RSp0334 contains PAS-PAC-GGDQF-EAL domains (Fig. 1b), we then continued to investigate its role in the metabolism of cyclic nucleotide molecules. High-performance liquid chromatography (HPLC) analysis showed that the deletion of *RSp0334* caused a substantial increase in the level of an unknown intracellular molecule, which was restored to the level of the wild-type strain by in trans expression of *RSp0334* (Supplementary Fig. 2a). As the retention time of this molecule was distinct from that of standard bis-3′,5′-c-di-GMP, the compound was isolated and purified from 300 L culture supernatant of *R. solanacearum* GMI1000. HR-ESI-MS analysis of the compound revealed a molecular ion [M-H]$^-$ with an m/z ratio of 344.03943, which suggested that the compound might have a molecular formula of $C_{10}H_{12}N_5O_7P$ (Supplementary Fig. 2b). On the basis of the HR-ESI-MS data, it was speculated that the isolated compound belongs to the cyclic GMP family instead of the cyclic di-GMP family. This speculation was further supported by the $^{31}P$ NMR data, in which there was a significant difference in the chemical shifts ($\delta$) of bis-3′,5′-c-di-GMP and the isolated compound. In details, the chemical shift ($\delta$) of the phosphine group in bis-3′,5′-c-di-GMP was −0.74, while the isolated compound has a chemical shift ($\delta$) of 20.01 (Supplementary Fig. 2c, d). Furthermore, $^1H$-$^1H$ COSY spectra showed that there was an obvious difference in the chemical shifts of the H-2 and H-3 of bis-3′,5′-c-di-GMP and the isolated compound (Supplementary Fig. 2e, f). Finally, by comparing the NMR information to that of 2′,3′-cGMP[24], we confirmed that the compound isolated from *R. solanacearum* was 2′,3′-cGMP (Supplementary Fig. 2g, i and Supplementary Table 1) and that the intracellular level of 2′,3′-cGMP was significantly increased in the *RSp0334* mutant (Supplementary Fig. 2j, k).

### RSp0334 catalyzes 2′,3′-cGMP to 2′,3′-c-di-GMP

Since RSp0334 negatively controlled the intracellular 2′,3′-cGMP level in *R. solanacearum* (Supplementary Fig. 2k), we hypothesized that it could use 2′,3′-cGMP as a substrate. To test this assumption, RSp0334, which has 894 amino acids and a calculated molecular weight of 96 kDa, was purified by using affinity chromatography and used for enzymatic activity experiments in vitro (Supplementary Fig. 3a). The results showed that more than 80% of 2′,3′-cGMP was catalyzed by the RSp0334 protein within 10 min (Supplementary Fig. 3b), while the protein did not catalyze or degrade 2′,3′-cyclic adenosine monophosphate (2′,3′-cAMP), GTP, bis-3′,5′-c-di-GMP or DNA in vitro (Supplementary Fig. 4). The product was then purified, and HR-ESI-MS analysis of the product revealed a molecular ion [M-H]$^-$ with an m/z ratio of 689.09003 (Supplementary Fig. 3c, d). In the $^{31}P$ NMR spectrum, it was observed that the product has two chemical shifts ($\delta$) of -1.20 and -1.75

(Supplementary Fig. 3e). These results suggested that the product is a cyclic di-GMP. Furthermore, the presence of two distinct peaks in the $^{31}$P NMR spectrum indicated that the product is an asymmetrical cyclic di-GMP, which was further supported by $^{1}$H NMR data. The $^{1}$H NMR spectrum revealed that peaks at the chemical shifts ($\delta$) 5.87, 5.82, 5.57, 4.34, 3.97, 3.54, and 3.43 were observed with an integration value of 1 (refer to Supplementary Fig. 3f), which also demonstrated that the product is an asymmetric as symmetric bis-3′,5′-c-di-GMP should have an even integration value. Therefore, we concluded that 2′,3′-cGMP was converted to 2′,3′-c-di-GMP by RSp0334 (Supplementary Fig. 3g-i). Our conclusion was further verified by triple-quadrupole MS in a multiple reaction monitoring model. The retention time and three characteristic mass transitions, $m/z$ 689/334(-) and $m/z$ 689/150(-) of the product confirmed it to be 2′,3′-c-di-GMP (Supplementary Fig. 3g, h). Consistently, the intracellular level of 2′,3′-c-di-GMP was significantly decreased in the *RSp0334* deletion mutant (Supplementary Fig. 3j). In addition, we purified two RSp0334 polypeptides containing only the GGDQF domain or EAL domain and tested their 2′,3′-cGMP catalysis activity (Supplementary Fig. 5a). HPLC analysis revealed that only the GGDQF domain catalyzed 2′,3′-cGMP to 2′,3′-c-di-GMP whereas the EAL domain of RSp0334 did not (Supplementary Fig. 5b). And in trans expression of the GGDQF domain almost completely restored the defective biofilm formation and motility phenotypes of the *RSp0334* mutant strain to those of the wild-type strain (Supplementary Fig. 5c, d) but did not affect the bacterial growth rates in different media (Supplementary Fig. 5e–g).

### The LLARLGGDQF motif catalyzes 2′,3′-cGMP to 2′,3′-c-di-GMP

As the GGDQF domain of RSp0334 catalyzes 2′,3′-cGMP to 2′,3′-c-di-GMP, while the GGDEF domain usually catalyzes GTP to bis-3′,5′-c-di-

GMP, we further investigated the detailed function of the GGDQF domain. To test whether the Q549 residue is responsible for the distinct functions between the GGDQF domain of RSp0334 and the GGDEF domain of bis-3′,5′-c-di-GMP synthase, we then generated a single point mutant, RSp0334 (GGDQF$^{Q549E}$), in which residue Q549 was substituted by Glu (E) to change the GGDQF domain to a GGDEF domain. Intriguingly, the results showed that RSp0334 (GGDQF$^{Q549E}$) still exhibited the same activity in catalyzing 2′,3′-cGMP but had no activity in converting GTP to bis-3′,5′-c-di-GMP (Supplementary Fig. 6a, b). To further explore the key enzyme active sites of the GGDQF domain, we then generated five mutant proteins with mutations of the five amino acid residues at G546, G547, D548, Q549, and F550, i.e., RSp0334 (GGDQF$^{AADQF}$), RSp0334 (GGDQF$^{GGAQF}$), RSp0334 (GGDQF$^{GGDAF}$), RSp0334 (GGDQF$^{GGDQA}$), and RSp0334 (GGDQF$^{AAAAA}$). All of the RSp0334 (GGDQF) derivatives were purified by using affinity chromatography and prepared for enzymatic activity experiments in vitro (Supplementary Fig. 6c). The HPLC analysis showed that only the mutation at Asp548 partially weakened the enzyme activity of RSp0334 (GGDQF), and mutation of all the five amino acid residues caused an obvious reduction of enzyme activity of RSp0334 GGDQF domain (Supplementary Fig. 6d–j and Supplementary Table 2). As expected, in trans expression of both *RSp0334(GGDQF$^{D548A}$)* and *RSp0334(AAAAA)* partially rescued the motility and biofilm formation of the *RSp0334* deletion mutant strain to the levels of the wild-type strain (Supplementary Fig. 6k, l).

To further determine whether other amino acid sites are required for the enzyme activity of RSp0334, we analyzed the amino acid alignment of the GGDQF domain of RSp0334 in *R. solanacearum* with the GGDEF domain (GGEEF) of WspR in *Pseudomonas aeruginosa* PAO1 and the GGDEF domain (GGEEF) of A1S_1695 in *Acinetobacter*

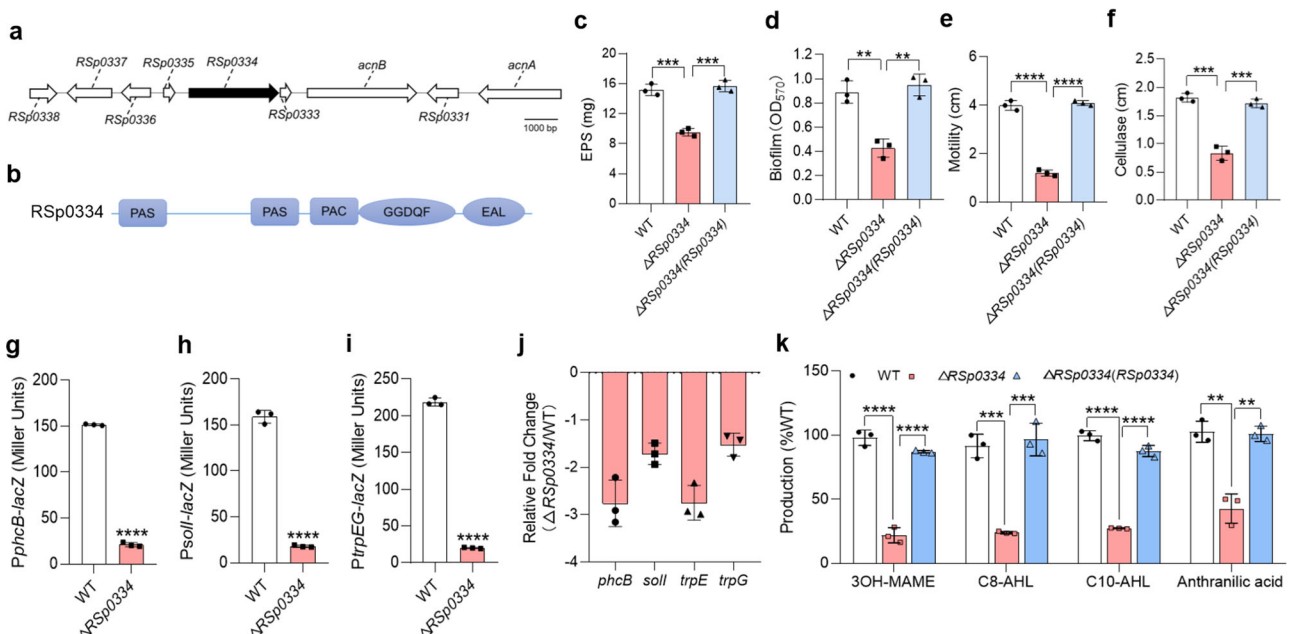

**Fig. 1 | Effects of RSp0334 on virulence-related phenotypes. a** Genomic organization of the *RSp0334* region in *R. solanacearum* GMI1000. **b** Domain structure analysis of RSp0334 in *R. solanacearum* (https://www.ebi.ac.uk/Tools/hmmer/). **c**–**f** The wild-type strain, the *RSp0334* mutant strain, and the *RSp0334* complemented strains were evaluated for the following virulence-related phenotypes: EPS production, biofilm formation, motility, and cellulase activity ($n = 3$ biological replicates). **c** ***$p = 0.0004$, ***$p = 0.0003$; **d** **$p = 0.0025$, **$p = 0.0015$; **e** ****$p < 0.0001$; **f** ***$p = 0.0003$, ***$p = 0.0005$. **g**–**i** Effects of RSp0334 on the gene expression levels of *phcB*, *soII* and *trpEG*, which were measured by assessing the β-galactosidase activity of *phcB-lacZ*, *soII-lacZ* and *trpEG-lacZ* transcriptional fusions in the wild-type and *RSp0334* mutant strains (OD$_{600}$ = 2.0) ($n = 3$ biological

replicates). **g** ****$p < 0.0001$; **h** ****$p < 0.0001$; **i** ****$p < 0.0001$. **j** The expression of signal synthase-encoding genes was evaluated by RT–qPCR in the wild-type and *RSp0334* mutant strains (OD$_{600}$ = 1.0) ($n = 3$ biological replicates). **k** The signal production in the wild-type, *RSp0334* mutant, and *RSp0334* complemented strains was analyzed with LC–MS ($n = 3$ biological replicates). The amounts of each signal in the *RSp0334* mutant and complement strains were normalized to that in the *R. solanacearum* wild-type strain, which was arbitrarily defined as 100%, ****$p < 0.0001$, ****$p < 0.0001$, ***$p = 0.0002$, ***$p = 0.0006$, ****$p < 0.0001$, ****$p < 0.0001$, **$p = 0.0018$, **$p = 0.0014$. Data are presented as mean ± SD and are representative of three independent experiments. The statistical comparisons were performed using one-way ANOVA. Source data are provided as a Source Data file.

*baumannii* ATCC 17978, both of which can catalyze the synthesis of bis-3',5'-c-di-GMP from GTP[17,25]. We found that RSp0334 possessed Leu542 and Leu545, which are different from the residues in the two bis-3',5'-c-di-GMP DGCs (Fig. 2a). Given that Asp548 contributes to the enzyme activity of RSp0334 (Supplementary Fig. 6d–j and Supplementary Table 2), we then generated the mutant protein RSp0334$^{L542A\ L545A\ D548A}$ (Fig. 2b) and tested its enzyme activity. The results showed that RSp0334$^{L542A\ L545A\ D548A}$ had no activity on 2',3'-cGMP (Fig. 2c), and *trans* expression of RSp0334$^{L542A\ L545A\ D548A}$ could not restore biofilm formation and motility of this mutant (Fig. 2d, e), suggesting that the three amino acid residues at L542, L545, and D548 are critical for the ability of RSp0334 to catalyze 2',3'-cGMP to 2',3'-c-di-GMP. More interestingly, a mutated WspR protein, WspR$^{A247L\ Y250L\ E253D}$, did not catalyze the synthesis of bis-3',5'-c-di-GMP from GTP but possessed the same 2',3'-cGMP to 2',3'-c-di-GMP catalysis activity as RSp0334 (Fig. 2f, g), and in trans expression of WspR$^{A247L\ Y250L\ E253D}$ could restore the motility and biofilm formation capacities of the *RSp0334* deletion mutant to 80% and 78.7% of wild-type strain levels, respectively (Fig. 2d, e). These observations indicate that the LLARLGGDQF motif catalyzes 2',3'-cGMP to 2',3'-c-di-GMP. We then further generated a mutant GGDQF domain protein of RSp0334 (LLARLGGDQF$^{L542A\ L545Y\ D548E}$) and found that the LLARLGGDQF$^{L542A\ L545Y\ D548E}$ variant had the same activity as WspR, synthesizing bis-3',5'-c-di-GMP from GTP. However, RSp0334 (LAARYGGDQF$^{LAARYAAAAA}$) did not catalyze the synthesis of bis-3',5'-c-di-GMP from GTP (Fig. 2h), suggesting that the amino acid residues at L542, L545 and D548 determine the ability of the RSp0334 enzyme to catalyze 2',3'-cGMP to 2',3'-c-di-GMP.

### RSp0980 is the receptor of 2',3'-cGMP that controls RSp0334-regulated phenotypes in *R. solanacearum*

As RSp0334 controls biological functions and modulates the intracellular levels of 2',3'-cGMP and 2',3'-c-di-GMP, we reasoned that there was likely to be an unknown effector that could sense 2',3'-cGMP, or 2',3'-c-di-GMP or both to control biological functions. As the structures of both 2',3'-cGMP and 2',3'-c-di-GMP are similar to that of bis-3',5'-c-di-GMP, we next searched for homologs of bis-3',5'-c-di-GMP effectors in *R. solanacearum* by using the BLAST program (https://blast.ncbi.nlm.nih.gov/Blast.cgi) to identify the potential effector of 2',3'-cGMP or 2',3'-c-di-GMP. We found 20 potential homologs sharing 25.18% to 50% identity with the identified bis-3',5'-c-di-GMP effectors or receptors (Supplementary Data 3). We then expressed each of the homologs in the *RSp0334* deletion mutant strain in trans and measured the motility activity of the transformed strains. We found that in trans expression of only *RSp0980* rescued the motility phenotype of the *RSp0334* mutant strain (Fig. 3a). Thus, *RSp0980* was selected for further investigation (Fig. 3b, c). Intriguingly, we found that in trans expression of *RSp0980* restored all the tested phenotypes of the *RSp0334* mutant strain to the wild-type strain levels, including EPS production, biofilm formation, and cellulase activity (Fig. 3d–f).

As in trans expression of *RSp0980* almost fully rescued the defective phenotypes of the *RSp0334* mutant strain, we continued to investigate the role of RSp0980 in regulating the important underlying biological functions. We generated an in-frame deletion of *RSp0980* and found that deletion of *RSp0980* caused the same phenotypic changes as were observed in the *RSp0334* mutant strain, including motility, biofilm formation, EPS production, and cellulase production (Fig. 3g–j), but did not affect the growth rates of the bacterial cells in either nutrient-rich or nutrient-poor medium (Supplementary Fig. 7).

Since RSp0980 regulates the same process as RSp0334 and has a REC domain, we hypothesized that it might sense 2',3'-cGMP, 2',3'-c-di-GMP or both. To test this hypothesis, we performed microscale thermophoresis (MST) analysis to test whether RSp0980 binds 2',3'-cGMP or 2',3'-c-di-GMP. RSp0980, which contained 222 aa with a calculated molecular weight of 25.6 kDa, was purified to homogeneity with affinity chromatography and then prepared for MST (Fig. 3k). As shown in Fig. 3l, RSp0980 bound strongly to 2',3'-cGMP, with an estimated dissociation constant ($K_D$) of $7.14 \pm 0.64\ \mu M$, whereas there was very weak binding between RSp0980 and 2',3'-c-di-GMP, with an estimated $K_D$ of $126.02 \pm 1.12\ \mu M$ (Fig. 3m). To further confirm the binding between RSp0980 and 2',3'-cGMP, we used isothermal titration calorimetry (ITC) analysis and the result showed that RSp0980 tightly bound to 2',3'-cGMP with an estimated dissociation constant ($K_D$) of $6.39 \pm 0.907\ \mu M$ (Fig. 3n). In addition, we purified two polypeptides containing only the HTH or REC domain of RSp0980 and found that the REC domain is the 2',3'-cGMP binding domain (Fig. 3o–q). Then, other cyclic nucleotide compounds were tested, and we found that RSp0980 did not bind to guanosine-2'- O-monophosphate (2'-GMP), guanosine-3'-O-monophosphate (3'-GMP), 3',5'-cyclic adenosine monophosphate (3',5'-cAMP), or 3',5'-cyclic guanosine monophosphate (3',5'-cGMP) but bound weakly to 2',3'-cAMP and bis-3',5'-c-di-GMP, with estimated $K_D$ values of $81.95 \pm 0.93\ \mu M$ and $130.37 \pm 1.3\ \mu M$, respectively (Supplementary Fig. 8a–f).

### RSp0980 regulates target gene expression by direct binding to promoter regions

The results showed that RSp0334 positively regulates the *phc, sol,* and anthranilic acid signaling systems in *R. solanacearum* (Fig. 1g–k). We then studied whether RSp0980 also regulates these systems. We constructed P*phcB-lacZ*, P*solI-lacZ,* and P*trpEG-lacZ* reporter systems in the *RSp0980* mutant strain, corresponding to the genes encoding the synthases of 3-OH MAME, AHL and anthranilic acid, respectively. Similar to the Δ*RSp0334* mutant, the *RSp0980* deletion resulted in reduced expression levels of *phcB, solI,* and *trpEG* (Fig. 4a–c). Therefore, we measured and compared the production of 3-OH MAME, C8-AHL, and C10-AHL and anthranilic acid signals in the wild-type, *RSp0980* mutant and complemented strains. The results showed that the production of 3-OH MAME, C8-AHL, C10-AHL, and anthranilic acid was reduced in the *RSp0980* mutant and that in trans expression of *RSp0980* almost fully rescued the level of signal production (Fig. 4d).

RSp0980 contains an HTH domain that is predicted to be closely involved in DNA binding (Fig. 3c). Therefore, we tested whether the transcriptional regulation of *phcB*, *solI* and *trpEG* is achieved by direct binding of RSp0980 to their promoters by performing electrophoretic mobility shift assays (EMSAs). The *phcB*, *solI*, and *trpEG* promoter DNA fragments formed stable DNA–protein complexes with RSp0980 and migrated slower than unbound probes. The amount of labeled probe bound to RSp0980 increased with increasing amounts of RSp0980 but decreased in the presence of a 20-fold greater concentration of the unlabeled probe (Fig. 4e-g). In line with the EPS production in the *RSp0980* mutant strain (Fig. 3h), RSp0980 regulated *epsA* expression by directly binding to its promoter (Fig. 4h). In addition, we purified both the HTH domain and REC domain of RSp0980 and found that the amounts of labeled probes bound to the HTH domain increased with increasing amounts of the HTH domain, while there was no DNA–protein complex formed between the probe and REC domain, suggesting that the HTH domain binds to the promoters of the target genes (Supplementary Fig. 9).

To further confirm that RSp0980 is a key downstream regulator of the RSp0334-mediated signaling system and elucidate the underlying regulatory mechanism, we employed a chromatin immunoprecipitation sequencing (ChIP-seq) assay to identify the target genes that were directly regulated by RSp0980 (Supplementary Data 4). Among the potential target genes, *phcB*, *solI*, *trpEG,* and *epsA* were associated with the biological functions and pathogenicity of *R. solanacearum*[20]. Sequence reads were obtained from ChIP-seq tests using an anti-His antibody and mapped to the *R. solanacearum* GMI1000 genome. The ChIP-seq data revealed that RSp0980 bound various gene promoters (Supplementary Data 4). Of the binding genes shown in Supplementary Data 4, *fliO* was chosen as an additional target gene for further analysis with EMSA. Similar to the above target genes, the amounts of

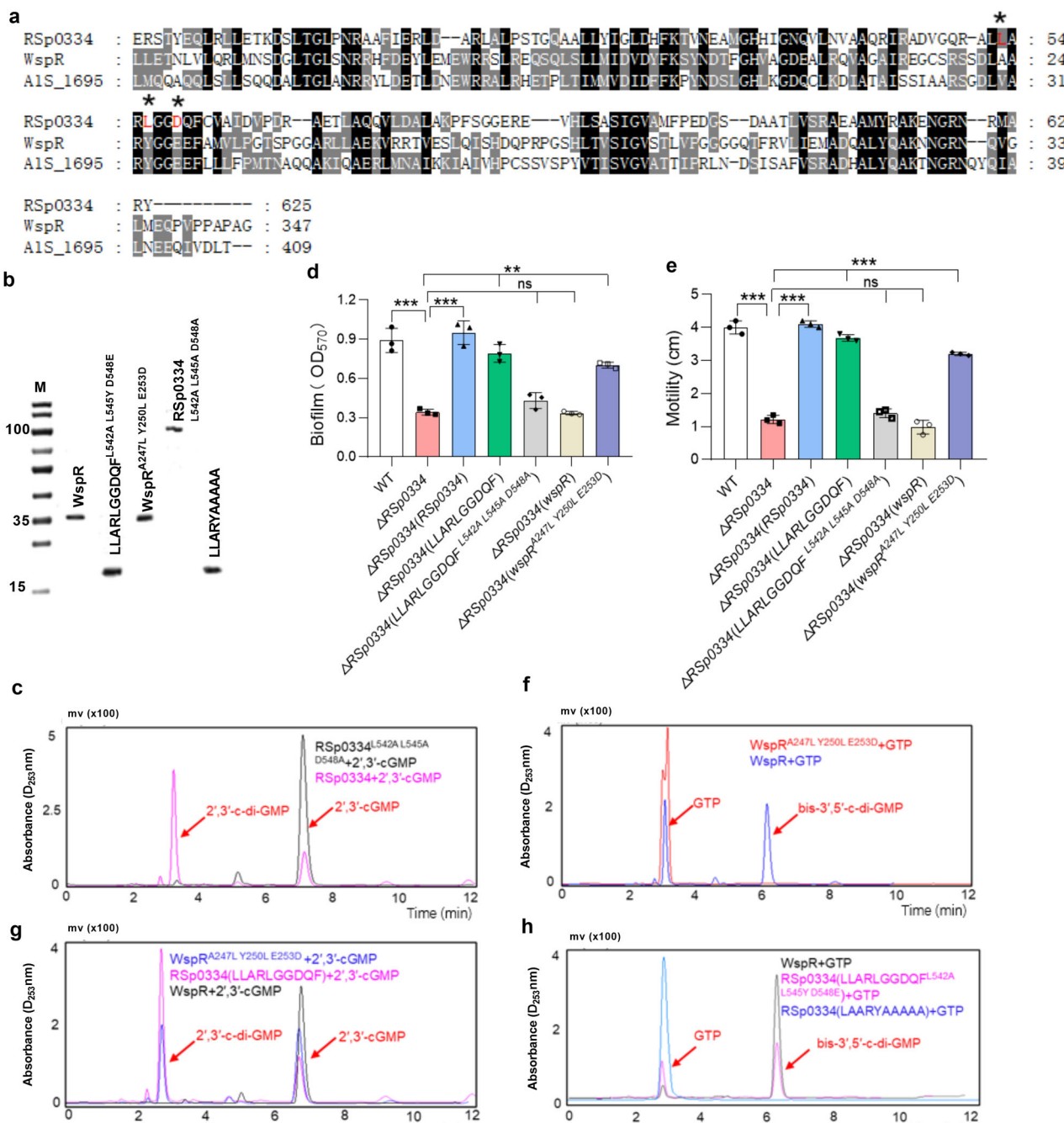

**Fig. 2 | Sequence analysis of the key enzyme active sites in RSp0334. a** Multiple sequence alignment of the GGDQF domain of RSp0334 in *R. solanacearum,* GGEEF domains of WspR in *P. aeruginosa* PAO1 and A1S_1695 in *A. baumannii* ATCC 17978 was performed with the ClustalW program and colored using GeneDoc 2.7. The key amino acid residues are marked in red and asterisk. **b** SDS–PAGE of the purified WspR, LLARLGGDQF$^{L542A\ L545Y\ D548E}$ motif of RSp0334, WspR$^{A247L\ Y250L\ E253D}$, RSp0334$^{L542A\ L545A\ D548A}$, RSp0334(LAARYAAAAA). **c** RSp0334 catalyzes 2′,3′-cGMP to 2′,3′-c-di-GMP, while RSp0334$^{L542A\ L545A\ D548A}$ showed no activity on 2′,3′-cGMP (RSp0334 + 2′,3′-cGMP: pink, RSp0334$^{L542A\ L545A\ D548A}$ + 2′,3′-cGMP: black). **d, e** In trans expression of *RSp0334*, *RSp0334(LLARLGGDQF)*, *wspR*, *wspR*$^{A247L\ Y250L\ E253D}$ and *RSp0334*$^{L542A\ L545A\ D548A}$ was evaluated for effects on the following virulence-related phenotypes: biofilm formation and motility (*n* = 3 biological replicates). **d** ***p* = 0.00057, ****p* = 0.00034, ****p* = 0.00039, $^{ns}$*p* = 0.0769, $^{ns}$*p* = 0.0621, *****p* < 0.0001; **e** *****p* < 0.0001, $^{ns}$*p* = 0.1569, $^{ns}$*p* = 0.1719. **f** WspR synthesizes bis-3′,5′-c-di-GMP from GTP, while WspR$^{A247L\ Y250L\ E253D}$ did not catalyze the production of

bis-3′,5′-c-di-GMP after mixing with GTP (WspR+GTP: blue, WspR$^{A247L\ Y250L\ E253D}$ + GTP: red). **g** WspR did not catalyze the production of 2′,3′-c-di-GMP after mixing with 2′,3′-cGMP, while WspR$^{A247L\ Y250L\ E253D}$ catalyzed 2′,3′-cGMP to 2′,3′-c-di-GMP, similar to the LLARLGGDQF motif of RSp0334 (WspR$^{A247L\ Y250L\ E253D}$ + 2′,3′-cGMP: blue, RSp0334(LLARLGGDQF) + 2′,3′-cGMP: pink, WspR+2′,3′-cGMP: black). **h** The mutated LLARLGGDQF$^{L542A\ L545A\ D548E}$ motif of RSp0334 was able to convert GTP to bis-3′,5′-c-di-GMP, similar to WspR, while RSp0334(LAARYAAAAA) showed no activity on GTP (WspR+GTP: black, RSp0334(LLARLGGDQF$^{L542A\ L545Y\ D548E}$ + GTP: pink, RSp0334(LAARYAAAAA) + GTP: blue). The retention time of 2′,3′-cGMP, 2′,3′-c-di-GMP, GTP and bis-3′,5′-c-di-GMP is 7 min, 3.5 min, 3.2 min and 6 min, respectively. Data are presented as mean ± SD and are representative of three independent experiments. In **b** experiment was performed three times and representative images from one experiment are shown. The statistical comparisons were performed using one-way ANOVA. Source data are provided as a Source Data file.

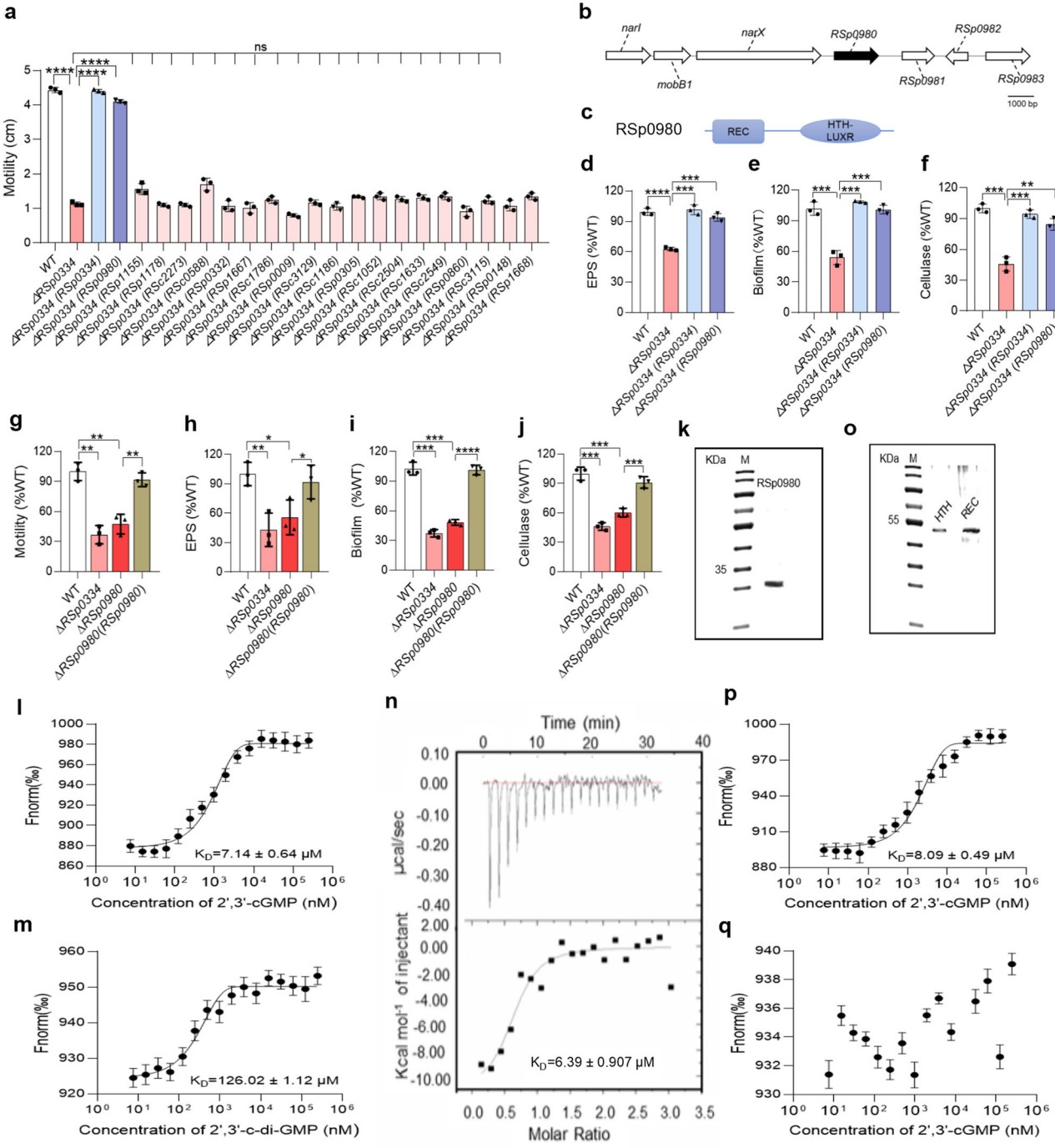

**Fig. 3 | Influence of *RSp0980* on virulence-related phenotypes in *RSp0334* mutant strains. a** Effects of the homologs of the bis-3′,5′-c-di-GMP receptor in *R. solanacearum* on the motility activity of *RSp0334*-deficient *R. solanacearum* GMI1000 (*n* = 3 biological replicates), *****p* < 0.0001, ⁿˢ*p* > 0.05. **b** Genomic organization of the *RSp0980* region in *R. solanacearum* GMI1000. **c** Domain structure analysis of RSp0980 in *R. solanacearum* (https://www.ebi.ac.uk/Tools/hmmer/). **d**–**f** In trans expression of *RSp0980* restored extracellular polysaccharide (EPS) production, biofilm formation, and cellulase activity in the *RSp0334*-deficient mutant strain (*n* = 3 biological replicates). **d** *****p* < 0.0001, ***p* = 0.00019, ***p* = 0.00019; **e** ***p* = 0.00095, ***p* = 0.00017, ***p* = 0.00056; **f** ***p* = 0.00032, ***p* = 0.00048, ***p* = 0.00017. **g**–**j** The wild-type strain, the *RSp0334* mutant strain, the *RSp0980* mutant strain and the *RSp0980* complemented strain were evaluated for the following virulence-related phenotypes: motility activity, EPS production, biofilm formation, and cellulase activity (*n* = 3 biological replicates). **g** ***p* = 0.00105, ***p* = 0.00244, ***p* = 0.00114; **h** ***p* = 0.00911, **p* = 0.0231,

**p* = 0.0252; **i** ****p* = 0.00011, ****p* = 0.00017, ****p* < 0.00001; **j** ****p* = 0.00026, ***p* = 0.001, ****p* = 0.00037; **k** SDS–PAGE of purified RSp0980. **l** Microscale thermophoresis (MST) analysis of the binding of RSp0980 to 2′,3′-cGMP (*n* = 4 biological replicates). **m** MST analysis of the binding of RSp0980 to 2′,3′-c-di-GMP (*n* = 4 biological replicates). **n** Isothermal titration calorimetry (ITC) analysis of the binding of RSp0980 to 2′,3′-cGMP. **o** SDS–PAGE of the purified REC domain and HTH domain of RSp0980. MST analysis of the binding of 2′,3′-cGMP to the REC domain **p** (*n* = 4 biological replicates) and HTH domain **q** (*n* = 4 biological replicates) of RSp0980. "Fnorm (‰)" indicates the fluorescence time trace changes in the MST response. Data are presented as mean ± SD and are representative of three independent experiments. The statistical comparisons were performed using one-way ANOVA or two-way ANOVA. In **k**–**o** and **n** experiment was performed three times and representative images from one experiment are shown. Source data are provided as a Source Data file.

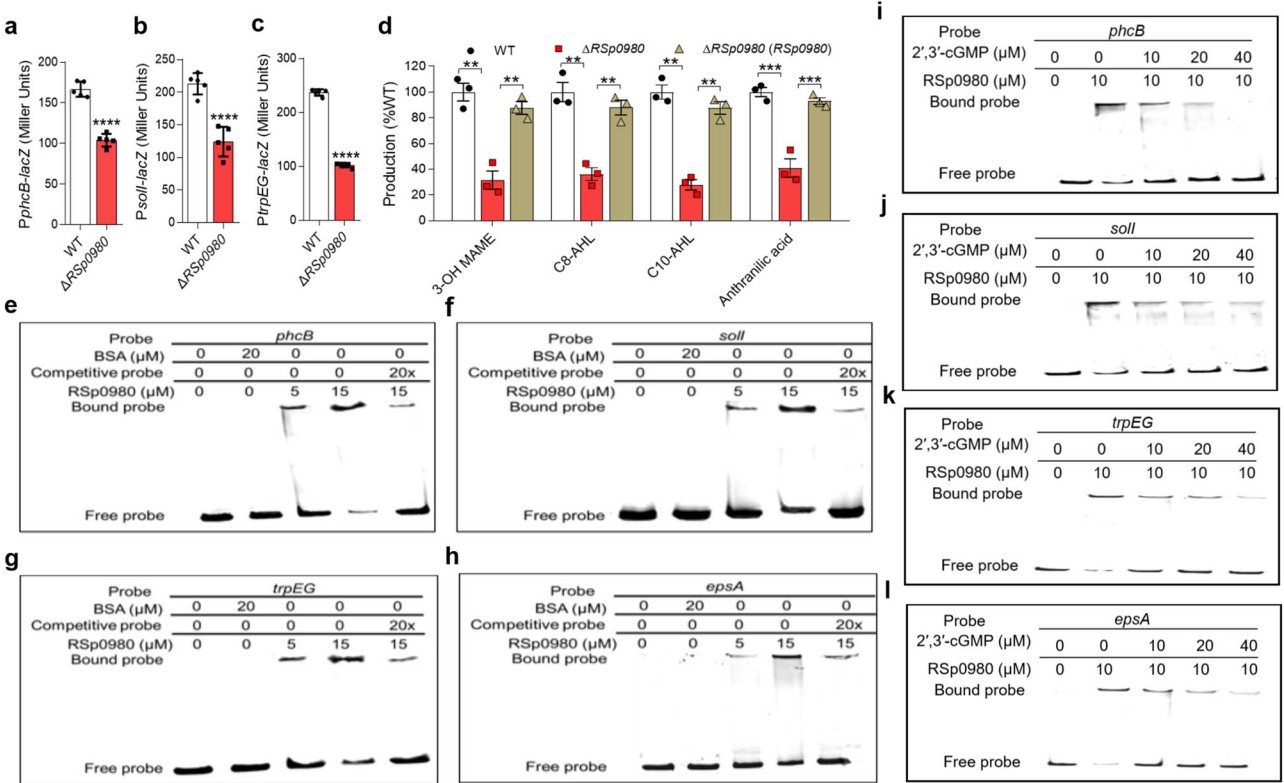

**Fig. 4 | Effects of RSp0980 on the QS signaling systems. a–c** Effects of *RSp0980* on the gene expression levels of *phcB*, *soII*, and *trpEG*, which were measured by assessing the β-galactosidase activity of the *phcB-lacZ*, *soII-lacZ* and *trpEG-lacZ* transcriptional fusions in the wild-type and *RSp0334* mutant strains (OD$_{600}$ = 2.0) (*n* = 5 biological replicates), respectively. **a** ****$p$ < 0.0001; **b** ****$p$ < 0.0001; **c** ****$p$ < 0.0001. **d** The QS signal production in *R. solanacearum* GMI1000 strains, which were analyzed by using LC–MS (*n* = 3 biological replicates). The amounts of each signal in the *RSp0980* mutant strain and complement strains were normalized to that in the *R. solanacearum* wild-type strain, which was arbitrarily defined as 100%, **$p$ = 0.00216, **$p$ = 0.0028, **$p$ = 0.00207, **$p$ = 0.00235, ***$p$ = 0.00045, ***$p$ = 0.00066, **$p$ = 0.00176, **$p$ = 0.00235. Data are presented as mean ± SD and

are representative of three independent experiments. The statistical comparisons were performed using one-way ANOVA. **e–h** Electrophoretic mobility shift assays (EMSAs) analysis of the binding of RSp0980 to the promoters of *phcB*, *soII*, *trpEG* and *epsA* in vitro. Biotin-labeled 327-bp *phcB*, 324-bp *soII*, 291-bp *trpEG* and 336-bp *epsA* promoter DNA probes were used for the protein binding assay. **i–l** EMSA detection of the in vitro binding of RSp0980 to the promoters of *phcB*, *soII*, *trpEG*, and *epsA* in the presence of different amounts of 2′,3′-cGMP. The protein was incubated with the probe in the presence of different concentrations of 2′,3′-cGMP at room temperature for 30 min. In **e–l** experiment was performed three times and representative images from one experiment are shown. Source data are provided as a Source Data file.

labeled *fliO* probe bound to RSp0980 also increased with increasing amounts of RSp0980 but decreased in the presence of a 20-fold greater concentration of the unlabeled probe (Supplementary Fig. 10a). HOMER (version 3) analysis of the ChIP-seq data revealed that these promoters contained a potential RSp0980-binding site, which was identified as 5′-GGAAATGGAC-3′ (Supplementary Fig. 10b). The potential binding sites 5′-CCTGCCCGAC-3′, 5′-GCAAATTCCG-3′, 5′-CGAACCCGAC-3′, and 5′-GGAAGTCGCC-3′ were then deleted from the promoter regions of *phcB*, *soII*, *trpEG*, and *epsA*, respectively. EMSA analysis showed that no DNA–protein complex was formed when the binding sites were deleted from the probes (Supplementary Fig. 10c–f), suggesting that these specific fragments are essential for the binding of RSp0980 to the promoters. Deletion of the potential binding site 5′-GGCGCTCGAC-3′ from the promoter of *fliO* also abolished its binding to RSp0980 (Supplementary Fig. 10g). Together, these results demonstrated that RSp0980 regulates target gene expression by directly binding to a specific region in the gene promoter.

## 2′,3′-cGMP abolishes the binding between RSp0980 and target promoter DNA

To determine how the binding of 2′,3′-cGMP to RSp0980 might affect the activity of RSp0980, we then examined the effects of 2′,3′-cGMP on the binding of RSp0980 to the promoters of *phcB*, *soII*, *trpEG* and *epsA*

by EMSA. As shown in Fig. 4i–l, the binding of RSp0980 to the *phcB*, *soII*, *trpEG*, and *epsA* promoter probes was inhibited when 2′,3′-cGMP was present in the reaction mixtures. The amounts of the probes that bound to RSp0980 decreased with increasing amounts of 2′,3′-cGMP. Consistent with the above results, the addition of exogenous 2′,3′-c-di-GMP or bis-3′,5′-c-di-GMP did not affect the binding of RSp0980 to the *epsA* promoter probe, while 2′,3′-cAMP slightly affected the binding of RSp0980 to the *epsA* promoter probe at 100 μM, suggesting that RSp0980 is a specific effector of 2′,3′-cGMP (Supplementary Fig. 8g–i).

## The RSp0334/2′,3′-cGMP/RSp0980 signaling system controls *R. solanacearum* pathogenicity and the expression of various genes

Previous studies have shown that mutants of *R. solanacearum* with defects in 3-OH MAME and anthranilic acid signaling are less virulent[20,26–28]. Therefore, we evaluated the effect of the RSp0334/2′,3′-cGMP/RSp0980 signaling system on the ability of *R. solanacearum* to infect host plants. As shown in Fig. 5a, compared with the plants infected by the *R. solanacearum* GMI1000 wild-type strain and complemented strains, those infected by the *RSp0334* mutant strain and *RSp0980* mutant strain exhibited significantly reduced wilting symptoms. At 21 d post inoculation, the disease indexes of tomato plants inoculated with the wild-type strain, *RSp0334* complement strains and *RSp0980* complement strains were 4.0, 3.8, and 3.8, respectively, while

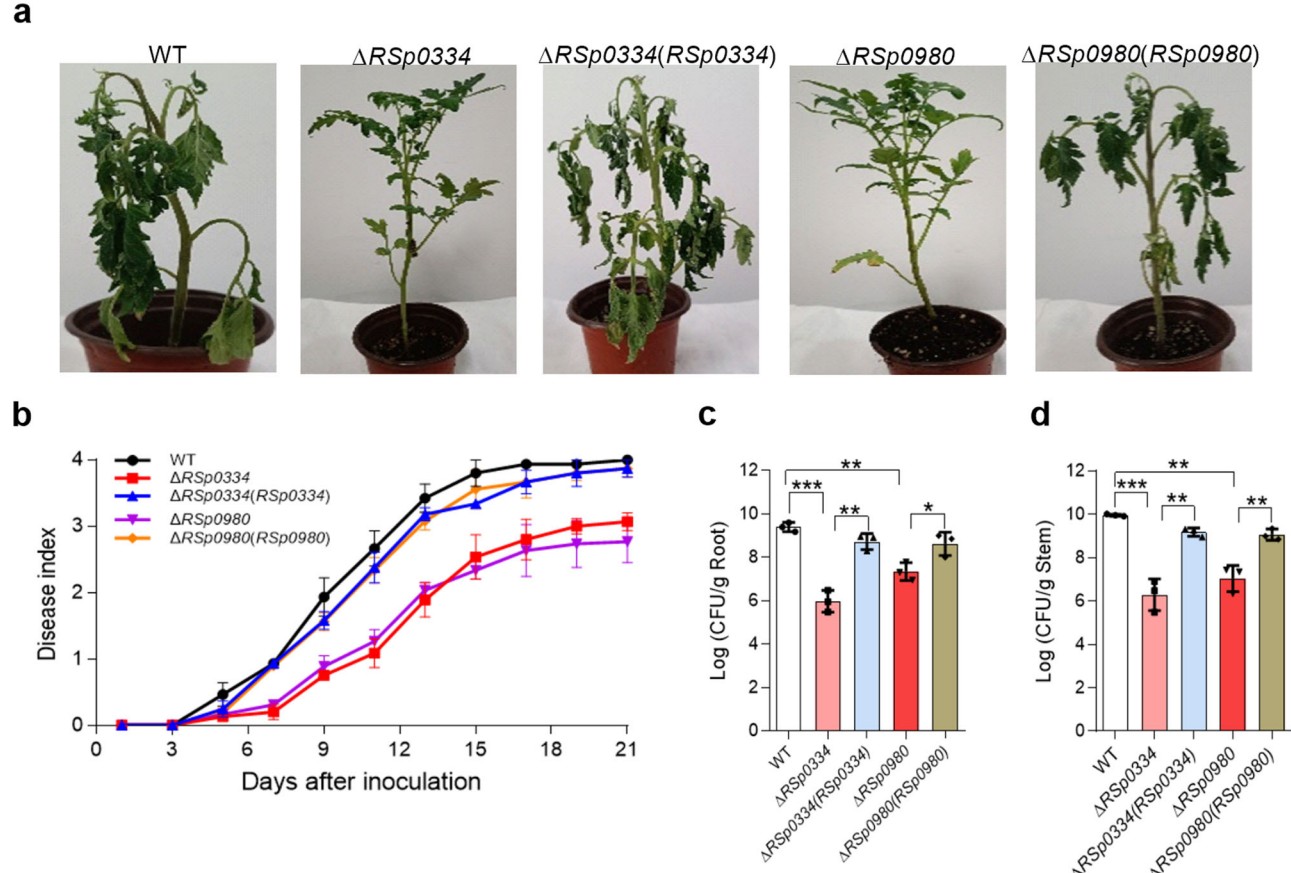

**Fig. 5 | Influence of *RSp0334* and *RSp0980* on the pathogenicity of *R. solanacearum* in tomato plants. a** Analysis of *R. solanacearum* virulence in tomato plants at 14 d after inoculation with *R. solanacearum*. **b** The effects of *RSp0334* and *RSp0980* on the virulence of *R. solanacearum*, measured by assessing the disease index of bacterial wilt in tomato (*n* = 7 biological replicates). **c, d** The number of colony-forming unit (CFU) of the *R. solanacearum* wild-type, *RSp0334* mutant, *RSp0980* mutant and complemented strains in the roots and stems of tomato plants (*n* = 3 biological replicates). Unwounded tomato plants were subjected to soil-soak inoculation with 8 mL of bacterial culture (OD$_{600}$ = 1.0) and incubated at 28 ± 1 °C under a 16-h light/8-hour dark cycle. **c** ****p* = 0.0004, ***p* = 0.00152, ***p* = 0.00159, **p* = 0.0807. **d** ****p* = 0.00097, ***p* = 0.00271, ***p* = 0.00112, ***p* = 0.00406. Data are presented as mean ± SD and are representative of three independent experiments. The statistical comparisons were performed using one-way ANOVA. Source data are provided as a Source Data file.

those of plants inoculated with the *RSp0334* mutant strain and *RSp0980* mutant strain were only 3.0 and 2.7, 25% and 32.5% lower than the wild-type strain, respectively (Fig. 5b). We then further quantified the colony-forming unit (CFU) values of these *R. solanacearum* strains in both the roots and stems of the tomato plants. The numbers of CFUs recorded for the *R. solanacearum* wild-type strain, *RSp0334* mutant strain, *RSp0980* mutant strain, *RSp0334* complemented strains and *RSp0980* complemented strains were $2.45 \times 10^9$, $9.32 \times 10^5$, $2.21 \times 10^7$, $5.18 \times 10^8$, and $4.12 \times 10^8$ per gram of root tissue at 7 d post inoculation, respectively (Fig. 5c). A similar result was observed for tomato stems, in which the numbers of CFUs of the five strains were $9.13 \times 10^9$, $1.93 \times 10^6$, $1.09 \times 10^7$, $1.51 \times 10^9$ and $1.16 \times 10^9$ per gram of stem tissue at 7 d post inoculation, respectively (Fig. 5d). These results suggested that the RSp0334/2′,3′-cGMP/RSp0980 signaling system plays an important role in the pathogenesis of *R. solanacearum*.

To determine the full scope of the regulatory role of 2′,3′-cGMP, we analyzed and compared the transcriptomes of the wild-type strain, *RSp0334* mutant strain and *RSp0980* mutant strain by using RNA sequencing (RNA-Seq). The expression levels of 126 genes were increased and those of 215 genes were decreased (log$_2$ fold-change ≥1.5) in the *RSp0334* mutant strain compared with the wild-type strain, whereas the expression levels of 65 genes were increased and those of 158 genes were decreased (log$_2$ fold-change ≥1.5) in the *RSp0980* mutant strain compared with the wild-type strain (Supplementary Fig. 11 and Supplementary Data 5). These differentially expressed genes were associated with a range of biological functions, including motility, virulence, and signal transduction (Supplementary Data 5). We also compared the transcriptomic profiles of the Δ*RSp0334* mutant and the Δ*RSp0980* mutant and found an overlap in their target genes, among them, 30 genes were upregulated and 100 genes were down-regulated (Supplementary Fig. 11e, f).

## Discussion

Bacteria employ a variety of mechanisms to sense and respond to environmental changes by converting extracellular signals or cues into intracellular signals[29]. Previous studies have demonstrated that intracellular secondary messengers, such as bis-3′,5′-c-di-GMP, regulate a variety of processes, including biofilm formation, motility, and pathogenicity in response to oxygen, nitric oxide, and a variety of other environmental challenges[2,13,17,30–33]. Additional nucleotide molecules, including 3′,5′-cyclic nucleotide monophosphate (3′,5′-cNMP), 3′,3′ cyclic GMP-AMP (3′,3′-cGAMP), 2′,3′,3′ cyclic AMP-AMP-AMP (2′,3′,3′-cAAA), 3′,3′,3′ cyclic AMP-AMP-GMP (3′,3′,3′-cAAG), and guanosine 5-diphosphate 3-diphosphate ((p)ppGpp), are also involved in the regulation of various bacterial physiological functions[34–39]. Recently, a class of nucleotide molecules, 2′,3′-cyclic nucleotide monophosphate (2′,3′-cNMP), were identified to play functional roles in both eukaryotes and prokaryotes[40–43]. In *E. coli*, 2′,3′-cNMP affect biofilm formation, motility, outer membrane structure and resistance to β-lactams and acid stress[44,45]. In this study, we identified 2′,3′-cGMP

as an intracellular second messenger that regulates diverse cellular processes in *R. solanacearum* (Fig. 1 and Supplementary Fig. 2). In addition, we found that the 2′,3′-cGMP signaling system controls the production of 3-OH MAME, AHL and anthranilic acid signals in *R. solanacearum* (Fig. 1 and Fig. 4). Our findings in this study establish the significant role of 2′,3′-cGMP in bacteria, and the importance of this second messenger will undoubtedly be further verified in the future.

Previous studies demonstrated that the GGDEF domains usually convert two GTPs to bis-3′,5′-c-di-GMP[46]. The object of this study, RSp0334, possesses PAS, PAC, GGDQF and EAL domains (Fig. 1b). We revealed that the GGDQF domain of RSp0334 can convert 2′,3′-cGMP to 2′,3′-c-di-GMP in the alkaline condition of pH 8.0 (Supplementary Fig. 5) and that RSp0334(GGDQF$^{Q549E}$) exhibits the same enzyme activity (Supplementary Fig. 6a). In addition, we found that Asp548 is involved in the enzyme activity of RSp0334 (Supplementary Fig. 6d–j and Supplementary Table 2) and that RSp0334 possessed Leu542 and Leu545, which are distinct from the corresponding residues in the two bis-3′,5′-c-di-GMP DGCs (Fig. 2a). RSp0334$^{L542A\ L545A\ D548A}$ showed no activity against 2′,3′-cGMP, whereas WspR$^{A247L\ Y250L\ E253D}$ did not catalyze the synthesis of bis-3′,5′-c-di-GMP from GTP but catalyzed 2′,3′-cGMP to 2′,3′-c-di-GMP (Fig. 2c–h), indicating that the LLARLGGDQF motif exerts a different function from the common GGDEF domain.

The main downstream functions of bis-3′,5′-c-di-GMP are achieved through the binding to its effectors and modulating their activities. It was demonstrated that 2′,3′-cNMP can bind to bacterial ribosomes and then result in decreased bacterial growth[47]. In this study, we identified that the transcriptional regulator RSp0980 is a receptor protein of 2′,3′-cGMP that controls the same biological functions as RSp0334 (Figs. 3–4). 2′,3′-cGMP was shown to bind to the REC domain of RSp0980 with high affinity and abolish the binding of RSp0980 to target promoter DNA (Fig. 3o–q and Fig. 4i–l). Moreover, RSp0980 showed no or very weak binding activity to 2′,3′-c-di-GMP, 2′,3′-cAMP, and bis-3′,5′-c-di-GMP (Supplementary Fig. 8a–f), indicating that RSp0980 is a specific receptor protein of 2′,3′-cGMP.

As RSp0334 converts two 2′,3′-cGMPs to one 2′,3′-c-di-GMP, which is an analog of the bacterial second messenger bis-3′,5′-c-di-GMP, it would be interesting to determine whether 2′,3′-c-di-GMP plays a role in regulation of bacterial physiology. However, as there is only very weak binding between RSp0980 and 2′,3′-c-di-GMP, and 2′,3′-c-di-GMP did not affect the binding of RSp0980 to the target promoter probes (Fig. 3m and Supplementary Fig. 8h), the potential functions and potential receptors of 2′,3′-c-di-GMP in *R. solanacearum* need to be further explored. In addition, bis-3′,5′-c-di-GMP is a potent immunostimulant in mammals that induces the production of type I interferons (IFNs) by directly binding to the endoplasmic reticulum-resident receptor stimulator of interferon genes (STING)[48]. A recent publication showed that 2′,3′-c-di-GMP acts as a potent agonist of STING in *Drosophila*, with functional relevance for the induction of antiviral immunity[49]. Therefore, it is also worth investigating the potential role of 2′,3′-c-di-GMP in bacteria.

As our study suggests that 2′,3′-cGMP is an important intracellular second messenger in *R. solanacearum*, it is worth to determine how this second messenger is synthesized. A recent study identified that 2′,3′-cGMP in plants is synthesized by TIR proteins[43]. In addition, mRNA turnover promoted by RNase I can produce 2′,3′-cNMPs in *E. coli* and *S. typhimurium*[42]. However, we have not found protein homologs of TIR and RNase I of *E. coli* and *S. typhimurium* in *R. solanacearum*. Interestingly, we found a homolog of ribonuclease T1 from *Cupriavidus plantarum* in *R. solanacearum* GMI1000. The in vitro enzyme experiment showed that the homologous protein RSc2766 catalyzes mRNA to 2′,3′-cGMP (Supplementary Fig. 12). We will also continue to study the role of RSc2766 in *R. solanacearum*. In addition, whether there are the similar 2′,3′-cGMP to 2′,3′-c-di-GMP pathways in eukaryotes is also worth exploring. Given that PDEs containing an HD, DHH or calcineurin-like PDE domain could hydrolyze 2′,3′-cGMP to 2′-GMP and

3′-GMP[50], our research may reveal a different conversion pathway for 2′,3′-cGMP.

In addition, we also revealed that *Salmonella typhimurium* ATCC14028, *Comamonas testosteroni* ATCC11996, *Burkholderia gladioli* ATCC10248, and *Bacillus amyloliquefaciens* AS1.210 produce 2′,3′-cGMP (Supplementary Fig. 13a–e). Intriguingly, it was found that the homolog of RSp0334 in *S. typhimurium* ATCC 14028, PdeR (UTL09452.1), possesses the LLARLGGDEF motif and exhibits the same enzyme activity as RSp0334, converting 2′,3′-cGMP to 2′,3′-c-di-GMP (Supplementary Fig. 13f). Moreover, the RSp0980 homolog in *S. typhimurium* ATCC14028, NarL (UTL09391.1), also specifically binds to 2′,3′-cGMP (Supplementary Fig. 13g–j). Together, our findings may trigger further investigations of the roles and mechanisms of this second messenger in diverse bacterial genera.

## Methods

### Bacterial strains and culture conditions

The bacterial strains and plasmids used in this study are listed in Supplementary Data 6. *R. solanacearum* GMI1000 (ATCCBAA-1114) was obtained from the American Type Culture Collection (ATCC). *R. solanacearum* strains were cultured in TTC medium (10 g/L tryptone (#LP0042B, oxoid), 5 g/L D-glucose (#G6152, Sigma-Aldrich), 1 g/L casein hydrolysate (#LP0041B, oxoid)) or on TTC agar (TTC medium containing 15 g/L agar (#A800728, Macklin)) at 28°C[20]. *E. coli* strains used for general cloning and conjugal transfer were cultured in Luria–Bertani medium (10 g/L tryptone, 5 g/L yeast extract (#PL0021, oxoid), 5 g/L NaCl (#S805275, Macklin), pH 7.0-7.5) or on LB agar (LB medium containing 15 g/L agar) at 37°C. In the EPS quantitative experiment, *R. solanacearum* was cultured in SP medium (5 g/L peptone (#P914715, Macklin), 20 g/L D-glucose, 0.5 g/L KH$_2$PO$_4$ (#P815662, Macklin) and 0.25 g/L MgSO$_4$ (#M813991, Macklin))[20]. 2′,3′-cGMP (#G025, Biolog), 3′,5′-cGMP (#G001, Biolog), 3′,5′-cAMP (#A001H, Biolog), 2′-GMP (#G022, Biolog), 3′-GMP (#G021, Biolog), GTP (#G810427, Macklin), GMP (#G822613, Macklin), 2′,3′-c-di-GMP (#tlrl-nacdg23, InvivoGen) and bis-3′,5′-c-di-GMP (#tlrl-nacdg, InvivoGen) was dissolved in H$_2$O to a final concentration of 1 mM. All media were supplemented with antibiotics according to the experimental requirements. The following antibiotics were used in this work: gentamicin (#G810322, Macklin), tetracycline (#T829835, Macklin), kanamycin (#K885955, Macklin), and ampicillin (#A830931, Macklin).

### Construction of in-frame deletion mutants and complementation

Gene knockout was achieved by homologous DNA recombination, and *R. solanacearum* GMI1000 was used as the parental strain[20]. For complementation analysis, the coding regions of genes sequences were amplified using 2×T5 Super PCR Mix (#TSE005, Tsinekg), then were cloned into the vector pLAFR3 under the control of the *lac* promoter (#C112-02, Vazyme) and introduced into the deletion mutant strains by using electroporation. The primers used for knockout, complementation and overexpression are listed in Supplementary Data 7.

### Extraction and identification of intracellular 2′,3′-cGMP

Bacteria were grown overnight in TTC medium at 28°C with shaking (OD$_{600}$ = 1.0). Formaldehyde (#F809902, Macklin, final concentration of 0.18%) was added to stop the degradation of 2′,3′-cGMP[23]. Cells were harvested by centrifugation (4000 x g) and washed 3 times with phosphate-buffered saline (PBS, pH 7.0) containing 0.18% formaldehyde. The cell pellets were dissolved in H$_2$O and boiled for 10 min. After cooling the samples on ice for 10 min, nucleotides were extracted in 65% ethanol (#809061, Macklin) in triplicate. The pooled supernatants were lyophilized, and the pellets were dissolved in 1 mL H$_2$O. Insoluble material was pelleted at 20,000 × g for 10 min. The resulting supernatant was filtered and performed with a Waters Xselect HSS T3 PREP (10 × 250 mm) followed by a Gemini-NX 5 u C18 110 A New

Column (250 × 10.0 mm) at a flow rate of 1 mL/min. The eluent for all analyses was a mixture of buffer A (0.01 M ammonium acetate (#A801000, Macklin) in water (pH 6.8)) and buffer B (methyl alcohol, #A452, ThermoFisher). NMR spectra were recorded at room temperature on a 400 MHz Bruker DPX 400. The residual solvent signals were taken as the reference (4.79 ppm for $^1$H NMR spectra $D_2O$ (#SJ22320024, Macklin)). The chemical shift ($\delta$) is reported in ppm, and coupling constants ($J$) are reported in Hz. The following abbreviations are used for multiplicity classifications: s = singlet, d = doublet, t = triplet, q = quartet, m = multiplet or unresolved, br = broad signal. Peptide masses were examined by using a ThermoFinnigan LCQ Fleet MS quadrupole ion trap mass spectrometer equipped with an ESI ion source. The production of 2′,3′-cGMP was measured by using LC–MS. The analyte detection was carried out on an API 3000 triple-quadrupole mass spectrometer equipped with an electrospray ionization source using selected reaction monitoring (SRM) analysis in negative ion mode.

## Construction of reporter strains and $\beta$-galactosidase activity measurement

We used promoter activity assays to quantify gene expression. The promoters of *phcB*, *solI*, *trpEG* and *epsA* were amplified using the primer pairs listed Supplementary Data 7. The 336-bp *epsA*, 291-bp *trpEG*, 327-bp *phcB*, and 324-bp *solI* promoter DNA products were inserted before the *lacZ* gene in the vector pME2-*lacZ*. Transconjugants were then selected on TTC agar plates supplemented with tetracycline and X-gal (#X8050, Solarbio). The *phcB*, *solI*, *trpEG*, and *epsA* reporter strains were cultured overnight in TTC medium supplemented with tetracycline at 28°C. Then, 200 μL of bacterial culture (OD$_{600}$ = 2.0) was centrifuged, and the cells were collected for determination of the $\beta$-galactosidase activity[21]. Each experiment was repeated three times in parallel.

## Quantitative analysis of QS signal production

*R. solanacearum* GMI1000 wild-type, *RSpO334* mutant and *RSpO980* mutant strains were grown in TTC medium overnight with agitation at 28°C. One liter of culture supernatant was collected by centrifugation (4000 x g) and extracted with an equal volume of ethyl acetate. The crude extract (organic phase) was dried by using a rotary evaporator and dissolved in methanol. All samples were kept at 4°C until analysis. Ultrahigh-performance liquid chromatography–electrospray ionization tandem mass spectrometry (UHPLC–ESI–MS/MS) was performed in a Shimadzu LC-30A UHPLC system with a Waters C18 column (1.8 μm, 150 × 2.1 mm) and a Shimadzu 8060 QQQ-MS mass spectrometer with an ESI source interface. The mass spectrometer was operated in negative-ion mode. The mobile phase was prepared as mixture of buffer A (0.01 M ammonium acetate in water (pH 6.8)) and buffer B (methyl alcohol).

## Protein purification and analysis

The coding region of each target gene was amplified with the primers listed in Supplementary Data 7 and fused to the expression vector PDBHT2 or pMAL-C5X. The resulting constructs were transformed into *E. coli* strain BL21 (DE3). Affinity purification of the fusion proteins was performed using HisTrap affinity columns (#SA035010, Smart-lifesciences) or maltose binding protein (MBP) affinity columns (#SA077010, Smart-lifesciences) according to the manufacturer's instructions. Fusion protein cleavage with TEV protease (#P2307, Beyotime) was conducted at 4°C overnight. The purified proteins were eluted and verified by SDS–PAGE[20].

## Enzyme activity assay

The assays were performed with 5 μM purified protein (final concentration) in a final volume of 45 μL of reaction buffer (50 mM Tris-HCl (pH 8.0), 300 mM NaCl, 10 mM MgCl$_2$). The reactions were initiated by adding substrate at a final concentration of 100 μM and allowed to proceed at 28°C for 10 min. Then, the reactions were stopped by heating for 10 min at 100°C. The products were analyzed by a Shimadzu LC-20AT high-performance liquid chromatography (HPLC) instrument equipped with a UV/Vis detector set to 253 nm. Separation was carried out using a reverse-phase T3 Waters column (2.1 × 40 mm; 5 μm) and a flow rate of 1 mL/min. Solvents containing 10% methanol ammonium acetate and 90% water were used. NMR spectra was recorded at room temperature on a 500 MHz Bruker DPX 500. The residual solvent signals were taken as the reference ($D_2O$ (#SJ22320024, Macklin)). The chemical shift ($\delta$) is reported in ppm, and coupling constants ($J$) are reported in Hz. The following abbreviations are used for multiplicity classifications: s = singlet, d = doublet, t = triplet, q = quartet, m = multiplet or unresolved, br = broad signal. Peptide masses were examined by using a ThermoFinnigan LCQ Fleet MS quadrupole ion trap mass spectrometer equipped with an ESI ion source. The LC-MS analyte detection was carried out on an API 3000 triple-quadrupole mass spectrometer equipped with an electrospray ionization source using selected reaction monitoring (SRM) analysis in negative ion mode[17]. MRM conditions were optimized using authentic standard chemicals, including 2′,3′-c-di-GMP ([M-H]$^-$ 689 > 334, 689 > 150).

## Electrophoretic mobility shift assay

EMSAs were performed according to the instructions for the Thermo Fisher Scientific kit with minor modifications[51]. In brief, the purified PCR products containing the promoters were 3-end labeled with biotin by using a Thermo Fisher Scientific Biotin 3′ End DNA Labeling Kit. Protein–DNA binding interactions were detected by using a LightShift Chemiluminescent EMSA Kit. DNA–protein binding reactions were performed according to the manufacturer's instructions (#20148, Thermo Fisher). A 5% polyacrylamide gel was used to separate the DNA–protein complexes. After UV crosslinking, the biotin-labeled probes in the membrane were detected using a biotin luminescence detection kit (#20148, Thermo Fisher).

## Virulence assay in tomato plants

Analysis of the virulence of the *R. solanacearum* wild-type strain, *RSpO334* mutant strain, *RSpO980* mutant strain, *RSpO334* complemented strains and *RSpO980* complemented strains was performed in an AIRKINS greenhouse (28°C, light 16 h and dark 8 h). A mixture of field soil, sand, and compost (1.25:1.25:0.5) was prepared and autoclaved at 121 °C for 20 min.

Tomato seeds (Jinfeng 1) were surface-sterilized in 2% NaClO (#MA-EN-EL-0P0716, Canrd) for 3 min and 75% ethanol for 2 min, rinsed 3 times in sterile water, and then planted in soil. The disease status of the tomato plants was assessed daily by scoring the disease index on a scale of 0 to 4 as previously described[20]. All plants were monitored for disease, and the following scale was used: 0, no symptoms; 1, 1–25% wilted leaves; 2, 26–50% wilted leaves; 3, 51–75% wilted leaves; and 4, 76-100% wilted leaves. At 7 d postinoculation, 1 g samples of plant root and stem tissue were collected, milled in a sterile mortar with 9 mL of sterile water and diluted in a gradient. Then, the diluted suspensions were plated on TTC plates to quantify the bacterial CFUs in the tomato roots and stems.

## Bacterial growth analysis

Overnight bacterial cultures in TTC medium were washed twice in fresh TTC medium, SP medium, or MP minimal medium and inoculated into the corresponding fresh medium to an optical density at 600 nm (OD$_{600}$) of 0.1[20]. Then, a 200 μL aliquot of cell suspension was added to each well at 28°C in a low-intensity shaking model using the Bioscreen-C automated growth curve analysis system. The medium was used as a blank control. MP minimal medium (1 L) was composed of the following constituents: FeSO$_4$·7H$_2$O (#F8263, Sigma-Aldrich),

1.25×10$^{-4}$ g; (NH$_4$)$_2$SO$_4$ (#A4418, Sigma-Aldrich), 0.5 g; MgSO$_4$·7H$_2$O (#V900270, Sigma-Aldrich), 0.05 g; and KH$_2$PO$_4$ (#P816385, Macklin), 3.4 g. The pH was adjusted to 7.0, and 20 mM glutamic acid (#G1251, Sigma-Aldrich) was added.

## Phenotypic analysis

Biofilm formation was quantified in polystyrene tissue culture plates (96-well)[20]. Overnight cultures were diluted to an OD$_{600}$ of 0.1 with TTC medium, aliquoted into 96-well polystyrene plates and grown in static culture for 20 h at 28°C. The culture medium was then removed from the wells, and the wells were stained with 0.1% crystal violet (#C6158, Sigma-Aldrich) for 15 min and washed 3 times with water before the addition of 75% ethanol. Biofilm formation was quantified by measuring the absorbance at 570 nm.

Motility activity was determined on semisolid agar (0.3%)[20]. Bacteria were inoculated into the centers of plates containing 1% tryptone and 0.3% agar. The plates were incubated at 28°C for 48 h, and then the colony diameter was measured.

Cellulase production was determined on carboxymethylcellulose sodium (CMS) solid medium (1 L contained 1 g of CMS (#C835846, Macklin), 3.8 g of Na$_3$PO$_4$ (#342483, Sigma-Aldrich), and 8.0 g of agar, pH 7.0)[21]. The CMS medium was added to a culture dish (F = 9 cm). An overnight culture of bacteria was diluted to an OD$_{600}$ of approximately 0.1, and 1 μL of the bacterial suspension was inoculated into the center of the CMS plates. The plates were incubated at 28°C for 48 h. The plates were stained with 0.5% Congo red (#IC1000, Solarbio) for 30 min. The plates were rinsed three times with 1 M NaCl, after which a transparent ring appeared; the size of the ring was measured as the indicator of cellulase activity.

EPS production was determined on SP medium[20]. First, 100 mL of overnight culture (OD$_{600}$ = 2.5) was centrifuged at 8000 × g for 20 min. The collected supernatants were mixed with a 4-fold volume of 95% ethanol, and the mixture was stored at 4 °C overnight. The precipitated EPS was isolated by centrifugation (8000 × g) and dried overnight at 55 °C before determination of the dry weight.

## Microscale thermophoresis assay

Protein binding experiments were carried out with a Nano Temper 16 Monolith NT.115 instrument (NanoTemper Technologies; www.nanotemper-technologies.com)[52]. In brief, the RSp0980 protein, HTH domain of RSp0980, and REC domain of RSp0980 were labeled with the L014 Monolith NT.115 Protein Labeling Kit (#MO-L018, Nano Temper). Labeled RSp0980 protein and the specified quantities of signaling molecules were mixed and loaded onto standard treated silicon capillaries (#MO-L022, Nano Temper), and fluorescence was measured. The measurements were carried out at 60% LED power and 40% MST power. Each combination in PBS buffer saline (#917808, Macklin) was tested at least three times.

## Isothermal titration calorimetry analysis

Isothermal titration calorimetry was used to characterize the binding affinity of 2′,3′-cGMP and RSp0980. The experiments were performed using an ITC-200 microcalorimeter following the manufacturer's protocol (MicroCal, Northampton)[23]. Both the 2′,3′-cGMP and Rsp0980 protein were dissolved Tris-buffer (20 mM Tris-HCl (#ST760, Beyotime), 100 mM NaCl, pH 8.5). Titrations began with one injection of 0.2 μL of 2′,3′-cGMP (200 μM) solution into the sample cell containing 350 μL of the protein solution (20 μM). The volume of the 2′,3′-cGMP injection was changed to 2 μL in the subsequent 19 injections. The heat changes accompanying the injections were recorded. The titration experiment was repeated at least three times, and the data were calibrated with the final injections and fitted to the one-site model to determine the binding constant ($K_D$) using MicroCal ORIGIN version 7 software.

## Real-time quantitative reverse transcription PCR

R. solanacearum cells were cultured to an OD$_{600}$ of 1.0 and then harvested. RNA was isolated using an Eastep Super Total RNA Extraction Kit (#LS1040, Promega). cDNA synthesis (#11141ES60, Yeasen) and RT–qPCR analysis was performed with SYBR qPCR Green Master Mix (#11201ES08, Yeasen) according to the manufacturer's protocol with LightCycler96 (Roche). The expression of the 16 S rRNA gene was used as the control. The relative expression levels of the target genes were calculated using the comparative CT ($2^{-\Delta\Delta CT}$) method[20].

## Chromatin immunoprecipitation-Seq

Chromatin immunoprecipitation assays were performed by Wuhan IGENEBOOK Biotechnology Co. Ltd.[53]. Briefly, 6 × 10$^{10}$ R. solanacearum cells were collected and cross-linked with 1% formaldehyde for 20 min at 37°C. The crosslinking was then stopped by the addition of glycine (#810676, Macklin) at a final concentration of 125 mM. Afterward, bacterial cells were centrifuged, washed twice with Tris buffer (150 mM NaCl, 20 mM Tris-HCl pH 7.5) containing a complete proteinase inhibitor cocktail (#4693116001, Roche), and then resuspended in 40 mL lysis buffer (50 mM Tris–HCl pH 8.0, 10 mM EDTA (#E8040, Solarbio), 1% SDS (#L5751, Lablead), 1% Triton X-100 (#T824275, Macklin), mini-protease inhibitor cocktail (#11836170001, Roche) for 30 min. The chromatin DNA was purified and sonicated to obtain soluble sheared chromatin with an average DNA length of 200–500 bp (20 s on with 30 s interval, 15 cycles, Diagenode Bioruptor Pico); 2 μL of chromatin was saved at -20°C as input DNA, and 100 μL of protein was used for immunoprecipitation with 5 μg of anti-His antibody (ab9108, 1:1000, Abcam) at 4°C overnight. The next day, 30 μL of protein G magnetic beads were added, and the reaction samples were further incubated at 4°C for 3 h. The beads were then washed with a series of washing buffers: once with low-salt washing buffer (20 mM Tris-HCl pH 8.1, 150 mM NaCl, 2 mM EDTA, 1% Triton X-100, 0.1% SDS); twice with LiCl washing buffer (10 mM Tris-HCl pH 8.1, 250 mM LiCl (#AM9480, ThermoFisher), 1 mM EDTA, 1% NP40 (#85124, ThermoFisher), 1% deoxycholic acid (#D8460, Solarbio)); and twice with TE buffer (10 mM Tris-HCl pH 7.5, 1 mM EDTA). Bound material was then eluted from the beads in 300 μL of elution buffer (100 mM NaHCO$_3$ (#S837271, Macklin), 1% SDS), treated first with RNase A (#EN0531, ThermoFisher) at a final concentration of 8 μg/mL for 6 h at 65°C and then with proteinase K (#39450-01-6, Macklin) at a final concentration of 345 μg/mL overnight at 45°C. Immunoprecipitated DNA was used to construct sequencing libraries following the protocol provided by the I NEXT-FLEX ChIP-Seq Library Prep Kit for Illumina Sequencing (NOVA-5143, Bioo Scientific) and sequenced on an Illumina Xten instrument with the PE 150 method.

Low-quality reads were filtered out via Trimmomatic software (version 0.38). In total, 46,400,552 and 45,766,286 clean reads were obtained from the input and ChIP samples, respectively. The clean reads were then mapped to the R. solanacearum GMI1000 genome with Bwa (v.0.7.15), allowing up to two mismatches. Samtools (v. 1.3.1) software was used to remove potential PCR duplicates. The software MACS2 (v.2.1.1.20160309) with default parameters (bandwidth, 300 bp/value, 0.05/model fold, 5, 50) was utilized to call peaks.

## RNA-seq analysis

Total RNA was isolated from R. solanacearum (OD$_{600}$ = 1.0) strains by using the Eastep Super Total RNA Extraction Kit. RNA purity was detected by using the NanoPhotometer® spectrophotometer (IMPLEN), RNA concentration was measured by using Qubit® RNA Assay Kit in Qubit® 2.0 Fluorometer (Life Technologies), and RNA integrity was assessed by using the RNA Nano 6000 Assay Kit with the Bioanalyzer 2100 system (Agilent Technologies). Sequencing libraries were generated by using the NEBNext® Ultra™ Directional RNA Library Prep Kit for Illumina® (NEB, USA). Finally, the products were purified (AMPure XP system), and library quality was assessed on the Agilent

Bioanalyzer 2100 system. The index-coded samples were clustered on a cBot Cluster Generation System using TruSeq PE Cluster Kit v3-cBot-HS (Illumina). After cluster generation, the library preparations were sequenced on an Illumina HiSeq platform, and paired-end reads were generated by Beijing Novogene Bioinformatics Technology Co., Ltd. The *R. solanacearum* GMI1000 reference genome and gene annotation files were downloaded from the genome website directly. Bowtie2-2.2.3 was used both to build the index of the reference genome and to align clean reads to the reference genome[54]. HTSeq v0.6.1 was used to count the read numbers mapped to each gene. Then, the FPKM of each gene was calculated based on the length of the gene and read count mapped to this gene[55].

## Statistical analysis

No statistical method was used to predetermine the sample size. No data were excluded from the analyses. Data are presented as the mean ± standard deviation (S.D.). Statistical analyses were performed using GraphPad Prism 9 software (Version 9.0.0). $p < 0.05$ was considered statistically significant. $*p < 0.05$; $**p < 0.01$; $***p < 0.001$; $****p < 0.0001$; ns, not significant. Biological replicates and numbers of independent experiments were stated in the legends. All experiments presented as representative HPLC graphs or gels were repeated at least 3 times with similar results.

## Reporting summary

Further information on research design is available in the Nature Portfolio Reporting Summary linked to this article.

## Data availability

RNA-seq and ChIP-seq data generated in this study have been deposited in the National Center for Biotechnology Information Sequence Read Archive (SRA) database under accession code PRJNA990578 and PRJNA990932, respectively. The data supporting the findings of this study are available within the article and Supplementary Information file. Source data are provided with this paper.

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

## Acknowledgements

This work was financially supported by the National Key Research and Development Program of China (2021YFA0717003 to Y.D.) and the Science, Technology and Innovation Commission of Shenzhen Municipality (JCYJ20200109142416497 to Y.D.), the National Natural Science Foundation of China (32300033 to S.S.), and Project Funded by China Postdoctoral Science Foundation (2022M713634 to S.S).

## Author contributions

Y.D. conceived the project. X.L., S.S., and Y.D. designed the research. X.L., W.Y., J.L., M.C., and S.S. performed the research. X.L., W.Y., J.L., Y.Z., Q.G., G.W., X.C., B.C., M.W., M.C, P.L., Y-W.H., W.Q., H.L., L-H.Z., X-W.L., S.S., and Y.D. analyzed the data. X.L., W.Y., J.L., S.S., and Y.D. wrote the paper. S.S. and Y.D. acquired funding.

## Competing interests

The authors declare no competing interests.
