## [Peer Review File · Nature Communications]

REVIEWER COMMENTS

Reviewer #1 (Remarks to the Author):

This manuscript by Li et. al. explores nucleotide signaling in the plant pathogen *Ralstonia solanacearum* (Rs). The authors conclude that a GGDEF/EAL-encoding enzyme in Rs, RSp0334, generates the novel 2'3'-c-di-GMP from two 2'3'-cGMP molecules. The synthase for 2'3'-cGMP is unknown. The authors then provide evidence that 2'3'-cGMP regulates many behaviors such as motility, biofilm formation, and plant infection by binding to and inhibiting the activity of the transcription factor RSp0980. The authors also provide evidence that 2'3'-cGMP signaling may be widespread. If correct, these findings would be highly significant and open up new areas in the field of cdN research. However, I do have some major concerns, primarily as stated in concern 1 it is not at all clear how a GGDEF enzyme could carry out this reaction. Furthermore, I did not find the mass spec data demonstrating the formation of 2'3'-c-di-GMP convincing as there seemed to be issues with the masses detected and a lack of chemically synthesized controls in each experiment.

1. My major concern with the conclusions of this manuscript regards the mechanism by which RSp0334 could generate 2'3'-c-di-GMP from two 2'3'-cGMP molecules. There are three families of cyclic dinucleotide (cdN) synthases: 1) diguanylate cyclase which synthesize c-di-GMP and in some cases c-GMP-AMP, 2) deadenylate cyclases which synthesize c-di-AMP, and CD-NTases, which synthesize a variety of cdNs. In all of these cases, the substrates for the synases are nucleotide tri-phosphates. Synthesis of the cdNs leads to a release of pyrophosphate molecules, which provides the energy to drive the synthesis reaction. In this manuscript, combining two substrate 2'3'-GMP would not allow the release of pyrophosphates, and thus it is not clear to me how the synthesis reaction would be driven forward. Furthermore, the authors shows that a "AAAAA" mutation of the putative GGDQF site still synthesizes 2'3'-c-di-GMP, which suggests that this enzyme is using a completely different active site. None of these issues are discussed by the authors.
2. Fig. 1J-The y-axis needs to define how the fold is being determined (Δ RSp0334/WT).
3. Line 128-A little detail in the text about how this molecule was purified is warranted.
4. Ext Data Fig. 2E-What are the higher MW peaks in the ESI-MS.
5. Ext Data Fig. 3E and 3F-I am confused by these data as they are supposed to be mass spectra of 2'3'-c-di-GMP. However, the molecular weight is 344.106 for both the standard and the reaction product. But the molecular weight of 3'3' cyclic di-GMP is 690.09, which would be 691.09 in positive ion mode I believe. So how can these molecules be 2'3'-c-di-GMP as they should be the same mass as 3'3'-c-di-GMP?
6. It is also a bit confusing that the 2'3'-cGMP in Ext Data Fig. 2B elutes at the same time as 2'3'-c-di-GMP in Figs. 2C and D. Are these the same HPLC system or different? The legend is unclear.
7. Ext Data Fig. 5D-Why is there a putative 2'3'-c-di-GMP peak for the 2'3'-cGMP standard?

8. In general, the HPLC experiments are not very convincing because they lack chemical standards. For example Ext Data Fig. 5D, there is no 2'3'-c-di-GMP alone peak. Likewise, In Ext Data Fig. 4C there is no 3'5'-c-di-GMP peak. Each of these experiments should have chemical standards for both the product and substrate.
9. I am confused by the nomenclature the authors are using. They conclude that Rsp0334 synthesizes 2'3'-c-di-GMP, which using the standard nomenclature for cyclic di-nucleotides, means that the phosphate ring consists of a 3'-5' and a 2'-5' bond. But then the authors refer to standard c-di-GMP as "3',5'". I am unsure what this means. I think it should be "3',3'" nomenclature.
10. Line 170-The authors need to specify here whether this is just the GGDEF domain or the entire protein. Based on Ext Data Fig. 6D it is the GGDEF domain.
11. Line 172-The authors should specify that the "AAAAA" mutant also had decreased activity.
12. Fig. 2D-There should be a WT WspR control in the experiment.
13. Fig. 2E-The authors argue that mutation of the L542, L545, and D548 residues in RSp0334 converts the enzyme to a standard diguanylate cyclase. The authors should make another mutation to inactivate the GGDEF active site to show this is required for this activity.
14. Line 192-The authors refer to the "LLARLGGDQF" novel motif that converts two 2'3'-cGMP to 2'3'-c-di-GMP, but their own mutation analysis indicates that "GG" and "QF" amino acids are not important for this activity and thus these should not be included as part of the domain.
15. None of the ChIP-seq data is shown so it is impossible to evaluate the listed targets.
16. Fig. 4I-J-The concentration of 2'3'-cGMP needed to reduce binding is very high, at 100 uM, even though the dissociation constant is 14. Even at these concentrations, there is still binding of O980 to the promoters. Therefore the evidence for 2'3'-cGMP inhibiting binding of O980 to the target promoters is not very strong.
17. Ext Data 11A-What does "TTC" stand for?
18. Lines 339-340-There is actually precedent for 2'3'-cNMPs in bacteria going back decades, and some nice recent work from the Weinert group that the authors should acknowledge and discuss. (PMID: 36439312, PMID: 34662237). One of these studies is very briefly mentioned in the discussion (line 386), but only near the end after the authors have stated multiple times that the discovery of 2'3'-cGMP in bacteria is novel.
19. More detail needs to be in the methods section regarding where the chemical standards for 2'3'-cGMP and 2'3'-c-di-GMP were obtained and how these were validated.

Reviewer #2 (Remarks to the Author):

I was very surprised that the authors claim that they identified a “new bacterial intracellular signal, 2',3'-cyclic guanosine monophosphate (2',3'-cGMP)”. In fact, 2',3'-cNMP has previously been proposed as a novel class of bacterial signals by the Weinert lab, see e.g.

Fontaine B.M., Martin K.S., Garcia-Rodriguez J.M., Jung C., Briggs L., Southwell J.E., Jia X., Weinert E.E. RNase I regulates *Escherichia coli* 2',3'-cyclic nucleotide monophosphate levels and biofilm formation. *Biochem. J.* 2018;475:1491–1506. doi: 10.1042/BCJ20170906.

Duggal Y., Kurasz J.E., Fontaine B.M., Marotta N.J., Chauhan S.S., Karls A.C., Weinert E.E. Cellular effects of 2',3'-cyclic nucleotide monophosphates in Gram-negative bacteria. *J. Bacteriol.* 2022;204:e0020821. doi: 10.1128/JB.00208-21.

Fontaine B.M., Duggal Y., Weinert E.E. 2',3'-Cyclic mononucleotide metabolism and possible roles in bacterial physiology. In: Chou S.H., Guiliani N., Lee V.T., Römling U., editors. *Microbial Cyclic Di-Nucleotide Signaling*. Springer Nature; Cham, Switzerland: 2020. pp. 627–637.

Chauhan S.S., Marotta N.J., Karls A.C., Weinert E.E. Binding of 2',3'-cyclic nucleotide monophosphates to bacterial ribosomes inhibits translation. *ACS Cent. Sci.* 2022;8:1518–1526. doi: 10.1021/acscentsci.2c00681.

Of these publications only the Duggal et al. paper is mentioned in the manuscript and this only in the context of 2',3'-cNMPs production, while this paper provides clear evidence that 2',3'-cNMPs signaling is widespread in bacteria (and eukaryotes), where it affects global gene expression, flagellar motility, biofilm formation and acid tolerance. Hence, not only the novelty of the data presented in this paper is limited it also fails to discuss the results in the light of the current state of knowledge.

It has been reported that in *E. coli* and *S. Typhimurium* 2',3'-cNMPs are generated through hydrolysis of RNA by RNAses. The important question of how this signal is produced in *Ralstonia* is not addressed in the present study. It rather identified an enzyme, RSp0334, which converts 2',3'-cGMP to 2',3'-c-di-GMP and thus inactivates the signal. However, the effect in vivo is not impressive, as inactivation of RSp0334 increased the cellular 2',3'-cGMP by about 50%. Notably, enzymes that hydrolyze 2',3'-cGMP such as 2',3'-cyclic phosphodiesterases have been extensively characterized in the enterobacteriaceae and have also been identified in eukaryotes. Overexpression of such enzymes was shown to fully deplete the cellular 2',3'-cNMP levels and were employed to investigate the functions of this signal in gene regulation. Are homologous enzymes present in *Ralstonia*? Do they play role in modulating 2',3'-cNMP levels? Why was the approach to enzymatically deplete 2',3'-cGMP not used to investigate the role of this signal in *Ralstonia*?

The paper then switches to 2',3'-cGMP perception and identified RSp0980 as a potential receptor for the molecule. However, the EMSA assays shown in Fig. 4 are unconvincing, the effect of 2',3'-cGMP is at the best very weak, the concentrations used appear very high. The proposed consensus sequence for RSp0980 binding is feeble. Moreover, RSp0980 is annotated as probable nitrate/nitrite response regulator. This predicted function of RSp0980 is not only supported by the fact the genes encoding the pathway for dissimilatory nitrate reduction is located next to the gene but also by the fact that homologs of RSp0980 have been demonstrated to be involved in nitrate sensing and expression of the dissimilatory nitrate reductase in other bacteria (see e.g. Mangalea MR, Borlee BR. *Sci Rep.* 2022 Jan 7;12(1):203. doi: 10.1038/s41598-021-04053-6. and Li W et al. *Int J Mol Sci.* 2022 Jun 29;23(13):7220. doi: 10.3390/ijms23137220). Finally, a recent study in *E. coli* showed that 2',3'-cNMPs bind to bacterial ribosomes, inhibit translation in vitro, and modulate growth rates, suggesting a potential mechanism for rapidly altering translation (Chauhan SS, Marotta NJ, Karls AC, Weinert EE. *ACS Cent Sci.* 2022 Nov 23;8(11):1518-1526. doi: 10.1021/acscentsci.2c00681). Such a mechanism that has a global effect on cellular physiology may also explain the phenotypes of the mutants reported in this study.

Reviewer #3 (Remarks to the Author):

Based on the unconventional behaviour of a GGDEF-EAL domain protein in the plant pathogen *Ralstonia solanacearum*, namely the downregulation of sessility equally as motility and virulence, and the subsequent high pressure liquid chromatography analysis inconsistent with the ubiquitous second messenger cyclic di-GMP to be regulated by this gene product, the authors initiated genetic, bioinformatic and biochemical assays which eventually led to the characterization of the RSp0334 GGDEF-EAL protein as a 2',3' cyclic di-GMP synthase using 2',3' cyclic GMP as a substrate and the identification of the response regulator RSp0980, a REC-HTH protein and homolog of the response regulator NarL of *Escherichia coli*, as a 2',3' cyclic GMP receptor. Identification of distinct amino acids in RSp0334 compared to unrelated diguanylate cyclase GGDEF domain proteins identified two leucine residues in close vicinity N-terminal of the GGDQF motif and the aspartate in the GGDQF motif required for processing of 2',3' cyclic GMP versus diguanylate cyclase activity in RSp0334 and the unrelated GGDEF diguanylate cyclase WspR. Transcriptome analyses showed that the differentially regulated genes compared to wild type partially overlap for RSp0334 and RSp0980. Chromatin immunoprecipitation assay identified promoter binding sites for RSp0980. Subsequently, the authors showed that 2',3' cyclic GMP inhibits the DNA binding activity of the RSp0980 transcriptional regulator by binding to its REC domain as demonstrated by electromobility shift assays using target promoters that encode key enzymes for three different classes of signaling molecules and an exopolysaccharide locus which are affected by 2',3' cyclic GMP.

General comments

This work shows an unexpected and intriguing functionality of a seemingly catalytically incompetent GGDEF domain and the power to follow up unconventional phenotypes of cyclic di-GMP signaling

proteins. The experimental design and follow up of experimentation is logic and easy to follow and most of the results shown are well done with a clear interpretation given by the authors. What would even more enrich the manuscript is molecular docking studies to suggest a binding site for 2',3' cyclic GMP on RSp0334 and RSp0980. I am also wondering where the energy for the synthesis of 2',3' cyclic di-GMP comes from, is the tension in the 2',3' cyclic GMP phosphate bond sufficient. Further I am wondering whether 2',3' cyclic GMP binds to the phosphorylated or unphosphorylated form of RSp0980. As an additional immediate follow up research questions I would consider the role of the EAL domain of Rsp0334. Is it catalytically functional?

Specific comments in consecutive order

I.35: ...RSp0980, a transcriptional regulator.

I.38: LLARLGGDQF is a motif. Also not sure I would call it a new domain as the functionalities between synthesis of 2',3' cyclic di-GMP and 3',3' cyclic di-GMP are easily interchangeable.

I.99: Is this identity given over the entire length of the protein? Needs to be clarified. If only GGDEF, EAL, GGDEF-EAL or PAS-GGDEF-EAL domains. Equally in Supplementary Table 2.

I.118: A short explanation to the genes and their products would be useful.

I.130: and elsewhere: In the light of the identity of the identified 2',3' cyclic GMP, similar information for 3',5'- cyclic GMP is relevant.

I.178: What has been the rationale to select WspR and A1S_1695 for comparison and not any other GGDEF diguanylate cyclase.

I.229: binding domain

I.323: Describe in more quantitative terms.

I.338: 3',5' cyclic GMP would be one of the closest controls.

Figure 2A: Highlight the relevant amino acids. Also, which part of the proteins is shown?

Figure 5A and Extended Data Fig. 6 E-J: Use the same maximum scale value on the y-axis for direct comparison.

I. 599: Describe the principle of construction of reporter plasmids. Which part of the promoter regions have been used?

Extended Data Fig. 2: Please show the control spectra for comparison.

Extended Data Fig. 3: Peaks in C and D show the same retention time.

Extended Data Fig. 4: Explain the different colours used for the graphs.

Extended Data Fig. 5D: Is the GGDQF domain a dimer?

Extended Data Fig. 10: mutated target gene promoter sequence or binding motif deleted?

Reviewer #4 (Remarks to the Author):

The authors have discovered a novel bacterial intracellular signaling mechanism that involves 2',3'-cyclic guanosine monophosphate (2',3'-cGMP), 2',3'-cyclic diguanosine monophosphate (2',3'-c-di-GMP) and two bacterial proteins, namely RSp0334 and RSp0980. Their results support the concept that in the plant pathogen *Ralstonia solanacearum* RSp0334 is an enzyme that converts 2',3'-cGMP to 2',3'-c-di-GMP, a reaction that lowers intracellular concentrations of 2',3'-cGMP while increasing intracellular concentrations of 2',3'-c-di-GMP. 2',3'-cGMP binds to its receptor RSp0980 and thereby prevents signaling by RSp0980. In the absence of 2',3'-cGMP, RSp0980 binds to the promoter region of a large number of genes and regulates cellular phenotype including virulence and quorum sensing signaling systems. Impairment of this signaling mechanism reduces the *R. solanacearum*-induced wilting of tomato plants. The authors provide evidence that this system exists also in *Salmonella typhimurium* (a human pathogen) and likely exists in a wide spectrum of bacteria. The RSp0334/2',3'-cGMP/RSp0980 pathway appears to be an important signaling mechanism in at least some species of bacteria. There are a few questions that should be addressed:

1. The K_m of RSp0334 for 2',3'-cGMP is approximately 100 μM and the K_d of 2',3'-cGMP for RSp0980 is approximately 10 μM . These data suggest that the intracellular concentrations of 2',3'-cGMP must be in the micromolar range to interact with these proteins. What is the intracellular level of 2',3'-cGMP in *Ralstonia solanacearum*? And how much do the intracellular concentrations of 2',3'-cGMP vary in bacteria? Are the levels within a reasonable range relative to the concentrations required to interact with the proposed components?
2. In the Discussion, the authors state that 3',5'-c-di-GMP is a potent immunostimulant in mammals that induces the production of type I interferons (IFNs) by directly binding to STING and therefore it would be worth investigating the potential role of 2',3'-c-di-GMP in the host (lines 378-381). InvivoGen sells 2',3'-c-di-GMP and the brochure provided by InvivoGen describes 2',3'-c-di-GMP as a potent stimulator of type I IFNs (even more potent than 3',5'-c-di-GMP). Apparently, the question as to whether 2',3'-c-di-GMP interacts with STING has already been answered. The authors should discuss this.
3. Recently, 2',3'-cGMP was discovered for the first time to exist in vivo and under physiological conditions in a mammalian species (*American Journal of Physiology-Regulatory, Integrative and Comparative Physiology* 316: R783–R790, 2019. PMC6620655, PMID: 30789788). In light of this information and the authors' current findings, the authors should discuss the possibility that 2',3'-cGMP may direct biochemical processes in mammals via a similar mechanism as described in bacteria.
4. Related to item #3, the AJP article mentioned above also shows that in vivo 2',3'-cyclic nucleotide 3'-phosphodiesterase (CNPase) metabolizes 2',3'-cGMP to 2'-GMP. Is there evidence for the existence of a homolog of the mammalian CNPase in bacteria? This could be a key player regulating 2',3'-cGMP levels.
5. Does *Ralstonia solanacearum* secrete 2',3'-cGMP or 3',5'-c-di-GMP into the culture medium? If so, this could provide a pathway for extracellular signaling. This should be discussed.
6. In the text and figures, the authors mention 2'-cGMP and 3'-cGMP. Don't you mean 2'-GMP and 3'-GMP? Not sure what the structure of 2'-cGMP and 3'-cGMP would be.

Reviewer #1 (Remarks to the Author):

This manuscript by Li *et al.* explores nucleotide signaling in the plant pathogen *Ralstonia solanacearum* (Rs). The authors conclude that a GGDEF/EAL-encoding enzyme in Rs, RSp0334, generates the novel 2'3'-c-di-GMP from two 2'3'-cGMP molecules. The synthase for 2'3'-cGMP is unknown. The authors then provide evidence that 2'3'-cGMP regulates many behaviors such as motility, biofilm formation, and plant infection by binding to and inhibiting the activity of the transcription factor RSp0980. The authors also provide evidence that 2'3'-cGMP signaling may be widespread. If correct, these findings would be highly significant and open up new areas in the field of cdN research. However, I do have some major concerns, primarily as stated in concern 1 it is not at all clear how a GGDEF enzyme could carry out this reaction. Furthermore, I did not find the mass spec data demonstrating the formation of 2'3'-c-di-GMP convincing as there seemed to be issues with the masses detected and a lack of chemically synthesized controls in each experiment.

1. My major concern with the conclusions of this manuscript regards the mechanism by which RSp0334 could generate 2'3'-c-di-GMP from two 2'3'-cGMP molecules. There are three families of cyclic di-nucleotide (cdN) synthases: 1) diguanylate cyclase which synthesize c-di-GMP and in some cases c-GMP-AMP, 2) deadenylate cyclases which synthesize c-di-AMP, and CD-NTases, which synthesize a variety of cdNs. In all of these cases, the substrates for the synthases are nucleotide tri-phosphates. Synthesis of the cdNs leads to a release of pyrophosphate molecules, which provides the energy to drive the synthesis reaction. In this manuscript, combining two substrate 2'3'-GMP would not allow the release of pyrophosphates, and thus it is not clear to me how the synthesis reaction would be driven forward. Furthermore, the authors shows that a "AAAAA" mutation of the putative GGDQF site still synthesizes 2'3'-c-di-GMP, which suggests that this enzyme is using a completely different active site. None of these issues are discussed by the authors.

Response: Thanks a lot for your valuable comments and suggestions. In the reaction conditions of RSp0334 to convert 2',3'-cGMP to 2',3'-c-di-GMP, the alkaline condition of pH 8.0 will trigger the hydrolysis of 2',3'-cGMP to 2'/3' GMP, which will then generate 2',3'-c-di-GMP by ion coordination. The product was then purified, and HR-ESI-MS analysis of the product revealed a molecular ion [M-H]⁻ with an m/z ratio of 689.09003 (Supplementary Fig. 3D). In the ³¹P NMR spectrum, it was observed that the product has two chemical shifts (δ) of -1.20 and -1.75 (Supplementary Fig. S3E). These results suggested that the product is a cyclic di-GMP. Furthermore, the presence of two distinct peaks in the ³¹P NMR spectrum indicated that the product is an asymmetrical cyclic di-GMP, which was further supported by ¹H NMR data. The ¹H NMR spectrum revealed that peaks at the chemical shifts (δ) 5.87, 5.82, 5.57, 4.34, 3.97, 3.54, and 3.43 were observed with an integration value of 1 (refer to Supplementary Fig. 3F), which also demonstrated that the product is an asymmetric as symmetric bis-3',5'-c-di-GMP should have an even integration value. Therefore, we concluded that 2',3'-cGMP was converted to 2',3'-c-di-GMP by RSp0334 (Supplementary Fig. 3G) (Line 152-162). Our conclusion was further verified by triple-quadrupole MS in a multiple reaction monitoring model. The retention time and three characteristic mass transitions, m/z 689/334(-) and m/z 689/150(-) of the product confirmed it to be 2',3'-c-di-GMP (Supplementary Fig. 3H-I) (Line 162-165).

For the GGDQF domain, we found that "AAAAA" mutation of this domain weakened the enzyme activity of RSp0334, and *in trans* expression of AAAAA only partially rescued the motility and biofilm formation of the RSp0334 deletion mutant strain to the levels of the wild-type strain, suggesting that the five amino acid residues are correlated with the active site (Supplementary Fig. 6), we have discussed this result in Line 190-192. In addition, we

also revealed that the mutant protein RSp0334^{L542A L545A D548A} showed no any enzyme activity on 2',3'-cGMP (Fig. 2C), and *trans* expression of RSp0334^{L542A L545A D548A} could not restore motility and biofilm formation capacity of this mutant (Fig. 2G-H), suggesting that the three amino acid residues at L542, L545, and D548 are critical for the ability of RSp0334 to catalyze 2',3'-cGMP to 2',3'-c-di-GMP. In combination, our results showed that L542, L545 and GGDQF domain contribute to the enzyme activity of RSp0334. We have described these details in Line 201-215.

1. Fig. 1J-The y-axis needs to define how the fold is being determined (deltaRSp0334/WT).

Response: Good suggestion, we have revised it as suggested.

2. Line 128-A little detail in the text about how this molecule was purified is warranted.

Response: Good suggestion, we have added more details in the text as suggested (Line 128-140).

3. Ext Data Fig. 2E-What are the higher MW peaks in the ESI-MS.

Response: We have repeated the experiment several more times and confirmed the highest peak (344.03943) in Supplementary Fig. 2B represents 2',3'-cGMP.

4. Ext Data Fig. 3E and 3F-I am confused by these data as they are supposed to be mass spectra of 2',3'-c-di-GMP. However, the molecular weight is 344.106 for both the standard and the reaction product. But the molecular weight of 3'3' cyclic di-GMP is 690.09, which would be 691.09 in positive ion mode I believe. So how can these molecules be 2',3'-c-di-GMP as they should be the same mass as 3'3'-c-di-GMP?

Response: The results in Extended Data Fig. 3E and 3F represent the high-resolution MS spectra of the standard 2',3'-c-di-GMP (Fig. S3E) and the RSp0334-generated products (Fig. S3F) in negative ion mode, respectively. The molecular weight of 2',3'-c-di-GMP is 690. Under ESI negative mode, both mass peaks of 689 and 344 will be observed. Typically, molecules will lose a proton under ESI negative mode, therefore, 2',3'-c-di-GMP will have a mass peak of 689. In addition, as 2',3'-c-di-GMP have 2 acid group (phosphate acid) that will lose two acid protons to resulting in a molecule of (M-2H)⁻ under ESI negative mode. As the mass spectrometer detects molecules by mass per charge ratio, which is (689 - 2)/2, thus gives a mass peak of 344. And we have repeated HR-ESI-MS analysis of the product several times and confirmed a molecular ion [M-H]⁻ with an m/z ratio of 689.09003 (Supplementary Fig. 3D).

5. It is also a bit confusing that the 2'3'-cGMP in Ext Data Fig. 2B elutes at the same time as 2'3'-c-di-GMP in Figs. 2C and D. Are these the same HPLC system or different? The legend is unclear.

Response: Good suggestion, the HPLC elution conditions for these reactions were the same, we have added more details in the figure legend as suggested.

The retention time of 2',3'-cGMP, 2',3'-c-di-GMP, GTP and bis-3',5'-c-di-GMP is 7 min, 3.5 min, 3.2 min and 6 min, respectively.

6. Ext Data Fig. 5D-Why is there a putative 2'3'-c-di-GMP peak for the 2'3'-cGMP standard?

Response: Thank a lot for your good question. We have found that the instrument contains a little residue of 2',3'-c-di-GMP sample. We have washed the machine several more times and repeated the HPLC experiments, and replaced the picture (Supplementary Fig. 5D).

1. In general, the HPLC experiments are not very convincing because they lack chemical standards. For example Ext Data Fig. 5D, there is no 2'3'-c-di-GMP alone peak. Likewise, In Ext Data Fig. 4C there is no 3'5'-c-di-GMP peak. Each of these experiments should have chemical standards for both the product and substrate.

Response: Good suggestion, we have added the standard chemicals as suggested.

2. I am confused by the nomenclature the authors are using. They conclude that Rsp0334 synthesizes 2'3'-c-di-GMP, which using the standard nomenclature for cyclic dinucleotides, means that the phosphate ring consists of a 3'-5' and a 2'-5' bond. But then the authors refer to standard c-di-GMP as "3',5'". I am unsure what this means. I think it should be "3',3'" nomenclature.

Response: Thanks a lot for your good question. We have modified the nomenclatures of the two chemicals. The two molecules were described as: bis-(3',5')-cyclic diguanosine monophosphate (abbreviated as bis-3',5'-c-di-GMP) and (2',5')(3',5')-cyclic diguanosine monophosphate (abbreviated as 2',3'-c-di-GMP) in Line 31 and Line 38, respectively.

3. Line 170-The authors need to specify here whether this is just the GGDEF domain or the entire protein. Based on Ext Data Fig. 6D it is the GGDEF domain.

Response: Good suggestion, we have added more details in the text as suggested.

4. Line 172-The authors should specify that the "AAAAA" mutant also had decreased activity.

Response: Good suggestion, we have added more details in the text as suggested. The HPLC analysis showed that only the mutation at Asp548 partially weakened the enzyme activity of RSp0334 (GGDQF), and mutation of all the five amino acid residues caused an obvious reduction of enzyme activity of RSp0334 GGDQF domain (Supplementary Fig. 6E-J and Supplementary Table 4) (Line 186-189).

5. Fig. 2D-There should be a WT WspR control in the experiment.

Response: Good suggestion, we have revised it as suggested. WspR did not catalyze the production of 2',3'-c-di-GMP after mixing with 2',3'-cGMP, while WspR^{A247L Y250L E253D} catalyzed 2',3'-cGMP to 2',3'-c-di-GMP, similar to the LLARLGGDQF domain of RSp0334 (WspR^{A247L Y250L E253D}+2',3'-cGMP: blue, RSp0334(LLARLGGDQF) +2',3'-cGMP: pink, WspR+2',3'-cGMP: black).

6. Fig. 2E-The authors argue that mutation of the L542, L545, and D548 residues in RSp0334 converts the enzyme to a standard diguanylate cyclase. The authors should make another mutation to inactivate the GGDEF active site to show this is required for this activity.

Response: In Fig. 2F, the mutated LLARLGGDQF^{L542A L545Y D548E} domain of RSp0334 was able to convert GTP to bis-3',5'-c-di-GMP, similar to WspR, while RSp0334(LAARYAAAAA) showed no activity on GTP (WspR+GTP: black, RSp0334(LLARLGGDQF^{L542A L545Y D548E}+GTP: pink, RSp0334(LAARYAAAAA)+GTP: blue).

7. Line 192-The authors refer to the "LLARLGGDQF" novel motif that converts two 2'3'-cGMP to 2'3'-c-di-GMP, but their own mutation analysis indicates that "GG" and "QF" amino acids are not important for this activity and thus these should not be included as part of the domain.

Response: Good suggestion. LLARLGGDQF is a region that catalyzes 2',3'-cGMP to 2',3'-c-di-GMP, we have revised the sentence as suggested.

8. None of the ChIP-seq data is shown so it is impossible to evaluate the listed targets.

Response: The ChIP-seq data revealed that RSp0980 bound various gene promoters (Supplementary Table 6) and the binding sequences of the proposed target gene promoters via ChIP-seq analysis were shown in Supplementary Fig.10B. And we have uploaded the ChIP-seq data to National Center for Biotechnology Information Sequence Read Archive (SRA) database under accession code PRJNA990932 according to the requirements of the Nature Communications Journal.

9. Fig. 4I-J-The concentration of 2',3'-cGMP needed to reduce binding is very high, at 100 μ M, even though the dissociation constant is 14. Even at these concentrations, there is still binding of 0980 to the promoters. Therefore the evidence for 2',3'-cGMP inhibiting binding of 0980 to the target promoters is not very strong.

Response: RSp0980 bound strongly to 2',3'-cGMP, with an estimated dissociation constant (K_D) of $7.14 \pm 0.64 \mu$ M (Line 245). We found that 2',3'-cGMP significantly inhibited RSp0980 from binding to the promoters of the target genes at 10 μ M, and we changed the concentrations of protein (RSp0980) and 2',3'-cGMP in the EMSA experiment, the results were shown as the following picture (Fig. 4).

10. Ext Data 11A-What does "TTC" stand for?

Response: TTC is a medium (Line 429), we have deleted it.

11. Lines 339-340-There is actually precedent for 2',3'-cNMPs in bacteria going back decades, and some nice recent work from the Weinert group that the authors should acknowledge and discuss. (PMID: 36439312, PMID: 34662237). One of these studies is very briefly mention in the discussion (line 386), but only near the end after the authors have stated multiple times that the discovery of 2',3'-cGMP in bacteria is novel.

Response: Good suggestion, we have added more details and cited these nice references as suggested in the discussion as suggested (Line 368-371).

12. More detail needs to be in the methods section regarding where the chemical standards for 2',3'-cGMP and 2',3'-c-di-GMP were obtained and how these were validated.

Response: Good suggestion, we have added more details in the methods as suggested (Line 434-436).

2',3'-cGMP, 3',5'-cGMP, 3',5'-cAMP, 2'-GMP, 3'-GMP, GTP, GMP, 2',3'-c-di-GMP and 3',5'-c-di-GMP (Biolog, California, America, HPLC \geq 95%) was dissolved in H₂O to a final concentration of 1 mM.

Reviewer #2 (Remarks to the Author):

I was very surprised that the authors claim that they identified a “new bacterial intracellular signal, 2',3'-cyclic guanosine monophosphate (2',3'-cGMP)”. In fact, 2',3'-cNMP has previously been proposed as a novel class of bacterial signals by the Weinert lab, see e.g.

Fontaine B.M., Martin K.S., Garcia-Rodriguez J.M., Jung C., Briggs L., Southwell J.E., Jia X., Weinert E.E. RNase I regulates *Escherichia coli* 2',3'-cyclic nucleotide monophosphate levels and biofilm formation. *Biochem. J.* 2018;475:1491–1506. doi: 10.1042/BCJ20170906.

Duggal Y., Kurasz J.E., Fontaine B.M., Marotta N.J., Chauhan S.S., Karls A.C., Weinert E.E. Cellular effects of 2',3'-cyclic nucleotide monophosphates in Gram-negative bacteria. *J. Bacteriol.* 2022;204:e0020821. doi: 10.1128/JB.00208-21.

Fontaine B.M., Duggal Y., Weinert E.E. 2',3'-Cyclic mononucleotide metabolism and possible roles in bacterial physiology. In: Chou S.H., Guilliani N., Lee V.T., Römling U., editors. *Microbial Cyclic Di-Nucleotide Signaling*. Springer Nature; Cham, Switzerland: 2020. pp. 627–637.

Chauhan S.S., Marotta N.J., Karls A.C., Weinert E.E. Binding of 2',3'-cyclic nucleotide monophosphates to bacterial ribosomes inhibits translation. *ACS Cent. Sci.* 2022;8:1518–1526. doi: 10.1021/acscentsci.2c00681.

Of these publications only the Duggal et al. paper is mentioned in the manuscript and this only in the context of 2',3'-cNMPs production, while this paper provides clear evidence that 2',3'-cNMPs signaling is widespread in bacteria (and eukaryotes), where it affects global gene expression, flagellar motility, biofilm formation and acid tolerance. Hence, not only the novelty of the data presented in this paper is limited it also fails to discuss the results in the light of the current state of knowledge.

It has been reported that in *E. coli* and *S. Typhimurium* 2',3'-cNMPs are generated through hydrolysis of RNA by RNAses. The important question of how this signal is produced in *Ralstonia* is not addressed in the present study. It rather identified an enzyme, RSp0334, which converts 2',3'-cGMP to 2',3'-c-di-GMP and thus inactivates the signal. However, the effect in vivo is not impressive, as inactivation of RSp0334 increased the cellular 2',3'-cGMP by about 50%. Notably, enzymes that hydrolyze 2',3'-cGMP such as 2',3' cyclic phosphodiesterases have been extensively characterized in the enterobacteriaceae and have also been identified in eukaryotes. Overexpression of such enzymes was shown to fully deplete the cellular 2',3'-cNMP levels and were employed to investigate the functions

of this signal in gene regulation. Are homologous enzymes present in *Ralstonia*? Do they play role in modulating 2',3'-cNMP levels? Why was the approach to enzymatically deplete 2',3'-cGMP not used to investigate the role of this signal in *Ralstonia*?

Response: Thanks a lot for your valuable comments and suggestions.

Firstly, in general, the novelties in our study are:

(1). We identified that the transcriptional regulator, RSp0980, is a novel receptor of 2',3'-cGMP; and 2',3'-cGMP specifically binds to RSp0980 with high affinity and thus abolishes the interaction between RSp0980 and the promoters of target genes.

(2). The LLARLGGDQF region of RSp0334 catalyzes 2',3'-cGMP to (2',5')(3',5')-cyclic diguanosine monophosphate (2',3'-c-di-GMP), and controls the important phenotypes through decreasing the intracellular 2',3'-cGMP levels.

(3). The LLD amino acid sites of RSp0334 determines the enzyme characteristics of RSp0334.

(4). The RSp0334/2',3'-cGMP/RSp0980 signaling system also exists in the human pathogen *Salmonella typhimurium*, and may exist in other bacteria.

We have added these papers mentioned above as references and discuss them in the discussion part in Line 368-371, Line 415-419

Secondly, we just detected the ratio of 2',3'-cGMP in the *RSp0334* deletion mutant strain to the wild-type strain at a time point, and we believe that this ratios between the two strains are dynamically changed at different time points.

Thirdly, we have listed all the homologs of the synthases and degrading enzymes of 2',3'-cNMP that have been identified so far, and we could not find any homolog of 2',3'-cGMP synthase in *Ralstonia solanacearum* GMI1000, suggesting that there might be a distinctive 2',3'-cGMP synthase in *R. solanacearum*, which is worth to be investigated in the future.

Table. Potential homologs of 2',3'-cNMP synthetase and hydrolase in *R. solanacearum*

Synthetase	Accession	Species	Reference	Identify
NLR immune receptors	7VU8_B	Nicotiana benthamiana	1	NA
RNase I	AAC73712	Escherichia coli	2	NA
Hydrolase	Accession	Species	Reference	Identify
tRNA-NT	P06961	Escherichia coli	3	53.68%
SpdA	WP_100669706	Sinorhizobium meliloti	4	33.33%
NUDT7	OAP01189	Arabidopsis thaliana	1	33.33%
PdeA	WP_201421553	Myxococcus xanthus	5	28.25%
PdeB	NOJ79288	Myxococcus xanthus	5	28%
YtqI	AAC00337	Bacillus subtilis	6	NA
CNPase	AH004086.2	Homo sapiens	7	NA

NA, not applicable

(1) Yu, D., et al., TIR domains of plant immune receptors are 2',3'-cAMP/cGMP synthetases mediating cell death. *Cell* **2022**, *185* (13), 2370-2386.

(2) Fontaine, B. M., et al., RNase I regulates *Escherichia coli* 2',3'-cyclic nucleotide monophosphate levels and biofilm formation. *Biochem. J.* **2018**, *475* (8), 1491-1506.

(3) Yakunin, A. F., et al., The HD domain of the *Escherichia coli* tRNA

nucleotidyltransferase has 2',3'-cyclic phosphodiesterase, 2'-nucleotidase, and phosphatase activities. *J. Biol. Chem.* **2004**, 279 (35), 36819-36827.

(4) Mathieu-Demaziere, C.; Poinso, V.; Masson-Boivin, C.; Garnerone, A. M.; Batut, J., Biochemical and functional characterization of SpdA, a 2', 3'-cyclic nucleotide phosphodiesterase from *Sinorhizobium meliloti*. *BMC Microbiol.* **2013**, 13, 268.

(5) Kimura, Y.; Okazaki, N.; Takegawa, K., Enzymatic characteristics of two novel *Myxococcus xanthus* enzymes, PdeA and PdeB, displaying 3',5'- and 2',3'-cAMP phosphodiesterase, and phosphatase activities. *FEBS Lett* **2009**, 583 (2), 443-8.

(6) Rao, F.; Qi, Y.; Murugan, E.; Pasunooti, S.; Ji, Q., 2',3'-cAMP hydrolysis by metal-dependent phosphodiesterases containing DHH, EAL, and HD domains is non-specific: Implications for PDE screening. *Biochem. Biophys. Res. Commun.* **2010**, 398 (3), 500-505.

(7) Thompson RJ. 2',3'-cyclic nucleotide-3'-phosphohydrolase and signal transduction in central nervous system myelin. *Biochem Soc Trans.* **1992** Aug;20(3):621-6.

The paper then switches to 2',3'-cGMP perception and identified RSp0980 as a potential receptor for the molecule. However, the EMSA assays shown in Fig. 4 are unconvincing, the effect of 2',3'-cGMP is at the best very weak, the concentrations used appear very high. The proposed consensus sequence for RSp0980 binding is feeble. Moreover, RSp0980 is annotated as probable nitrate/nitrite response regulator. This predicted function of RSp0980 is not only supported by the fact the genes encoding the pathway for dissimilatory nitrate reduction is located next to the gene but also by the fact that homologs of RSp0980 have been demonstrated to be involved in nitrate sensing and expression of the dissimilatory nitrate reductase in other bacteria (see e.g. Mangalea MR, Borlee BR. *Sci Rep.* 2022 Jan 7;12(1):203. doi: 10.1038/s41598-021-04053-6. and Li W et al. *Int J Mol Sci.* 2022 Jun 29;23(13):7220. doi: 10.3390/ijms23137220). Finally, a recent study in *E. coli* showed that 2',3'-cNMPs bind to bacterial ribosomes, inhibit translation in vitro, and modulate growth rates, suggesting a potential mechanism for rapidly altering translation (Chauhan SS, Marotta NJ, Karls AC, Weinert EE. *ACS Cent Sci.* 2022 Nov 23;8(11):1518-1526. doi: 10.1021/acscentsci.2c00681). Such a mechanism that has a global effect on cellular physiology may also explain the phenotypes of the mutants reported in this study.

Response: Thanks a lot for your valuable comments and suggestions.

Firstly, RSp0980 bound strongly to 2',3'-cGMP, with an estimated dissociation constant (K_D) of $7.14 \pm 0.64 \mu\text{M}$ (Line 245), 2',3'-cGMP also significantly inhibited RSp0980 from binding to the promoter of the target gene at $10 \mu\text{M}$ (Fig. 4).

Secondly, we examined the effect of nitrate on the phenotypes of *R. solanacearum* and we found that the exogenous addition of NaNO_3 at low concentration didn't affect the biofilm formation and motility of the wild-type strain, but only addition of high concentrations of NaNO_3 at 1mM and 10 mM obviously inhibited the biofilm formation and motility of *R. solanacearum* wild-type strain. Interestingly, exogenous addition of NaNO_3 did not affect the phenotypes of the *RSp0980* deletion mutant, suggesting that RSp0980 is possible to be involved in nitrate sensing, we have added the discussion and relevant references in Line 399-401.

Finally, previous study showed that 2',3'-cNMPs alter bacterial growth by bind to bacterial ribosomes ((Chauhan SS, Marotta NJ, Karls AC, Weinert EE. ACS Cent Sci. 2022 Nov 23;8(11):1518-1526). However, the deletion of *RSp0334* had little effect on the growth of bacterial cells in different media (Supplementary Fig. 1C-E) but resulted in significant defects in phenotypes including motility, biofilm formation, cellulase production, and extracellular polysaccharide (EPS) production, and *in trans* expression of *RSp0334* restored all of the phenotypes of the *RSp0334* deletion mutant to those of the wild-type strain (Fig. 1C-F) (Line107-111), suggesting that 2',3'-cGMP utilized a different mechanism in *R. solanacearum* from that in *E. coli*.

Reviewer #3 (Remarks to the Author):

Based on the unconventional behaviour of a GGDEF-EAL domain protein in the plant pathogen *Ralstonia solanacearum*, namely the downregulation of sessility equally as motility and virulence, and the subsequent high pressure liquid chromatography analysis inconsistent with the ubiquitous second messenger cyclic di-GMP to be regulated by this gene product, the authors initiated genetic, bioinformatic and biochemical assays which eventually led to the characterization of the RSp0334 GGDEF-EAL protein as a 2',3' cyclic di-GMP synthase using 2',3' cyclic GMP as a substrate and the identification of the response regulator RSp0980, a REC-HTH protein and homolog of the response regulator NarL of *Escherichia coli*, as a 2',3' cyclic GMP receptor. Identification of distinct amino acids in RSP0334 compared to unrelated diguanylate cyclase GGDEF domain proteins identified two leucine residues in close vicinity N-terminal of the GGDQF motif and the aspartate in the GGDQF motif required for processing of 2',3' cyclic GMP versus diguanylate cyclase activity in RSp0334 and the unrelated GGDEF diguanylate cyclase WspR. Transcriptome analyses showed that the differentially regulated genes compared to wild type partially overlap for RSp0334 and RSp0980. Chromatin immunoprecipitation assay identified promoter binding sites for RSp0980. Subsequently, the authors showed that 2',3' cyclic GMP inhibits the DNA binding activity of the RSp0980 transcriptional regulator by binding to its REC domain as demonstrated by electromobility shift assays using target promoters that encode key enzymes for three different classes of signaling molecules and an exopolysaccharide locus which are affected by 2',3' cyclic GMP.

General comments

This work shows an unexpected and intriguing functionality of a seemingly catalytically incompetent GGDEF domain and the power to follow up unconventional phenotypes of cyclic di-GMP signaling proteins. The experimental design and follow up of experimentation is logic and easy to follow and most of the results shown are well done with a clear interpretation given by the authors. What would even more enrich the manuscript is molecular docking studies to suggest a binding site for 2',3' cyclic GMP on RSp0334 and RSp0980. I am also wondering where the energy for the synthesis of 2',3' cyclic di-GMP comes from, is the tension in the 2',3' cyclic GMP phosphate bond sufficient. Further I am wondering whether 2',3' cyclic GMP binds to the phosphorylated or unphosphorylated form of RSp0980. As an additional immediate follow up research questions I would consider the role of the EAL domain of Rsp0334. Is it catalytically functional?

Response: Thank you very much for your kind and valuable comments.

Firstly, in our study, we have used an alkaline condition of pH 8.0. 2',3'-cGMP will be hydrolyzed to 2'/3' GMP in these conditions, and then they will generate (2',5')(3',5')-c-di-GMP (2',3'-c-di-GMP) by ion coordination. In addition, the 2',3'-cGMP was converted to 2',3'-c-di-GMP by RSp0334 within 10 min (Supplementary Fig. 3B-I), while RSp0334 did not catalyze or degrade 2',3'-cAMP, GTP, bis-3',5'-c-di-GMP or DNA *in vitro* (Supplementary Fig. 4) (Line 150-152), suggesting that 2',3'-cGMP is a specific substrate of RSp0334.

Secondly, we have detected the phosphorylation sites of RSp0980 by using mass spectrometry as suggested. The result showed that Thr amino acid at 12 residue was phosphorylated. The result was shown as follows.

MT(+79.97)IRILLIDDHT(+79.97)LFRSGIR

Thirdly, we purified two RSp0334 polypeptides containing only the GGDQF domain or EAL domain and tested their 2',3'-cGMP catalysis activity (Supplementary Fig. 5A). HPLC analysis revealed that only the GGDQF domain catalyzed 2',3'-cGMP to 2',3'-c-di-GMP whereas the EAL domain of RSp0334 did not (Supplementary Fig. 5D) (Line 165-168). Furthermore, the EAL domain of RSp0334 did not degrade bis-3',5'-c-di-GMP *in vitro*.

Specific comments in consecutive order

I.35: ...RSp0980, a transcriptional regulator.

Response: Good suggestion, we have revised it as suggested.

I.38: LLARLGGDQF is a motif. Also not sure I would call it a new domain as the functionalities between synthesis of 2',3' cyclic di-GMP and 3',3' cyclic di-GMP are easily interchangeable.

Response: Thanks a lot for your valuable comments and suggestions. We have revised LLARLGGDQF as a region that catalyzes 2',3'-cGMP to 2',3'-c-di-GMP (Line 38, 82, 173, 209, 387).

I.99: Is this identity given over the entire length of the protein? Needs to be clarified. If only GGDEF, EAL, GGDEF-EAL or PAS-GGDEF-EAL domains. Equally in Supplementary Table 2.

Response: Good suggestion, we have added more details in text and Supplementary Table 2.

I.118: A short explanation to the genes and their products would be useful.

Response: Good suggestion, we have added more details in Line 113, Line 118.

As the tested phenotypes are controlled by the anthranilic acid signaling and 3-hydroxypalmitic acid methyl ester (3-OH MAME) QS systems, we then tested whether there is a relationship between RSp0334 and these signaling systems.

Consistent with these results, the levels of 3-OH MAME, N-octanoyl-L-homoserine lactone (C8-AHL), N-decanoyl-L-homoserine lactone (C10-AHL) and anthranilic acid production were reduced in the *RSp0334* mutant strain by 78.13%, 76.08%, 72.74% and 57.27%, respectively, and *in trans* expression of *RSp0334* almost completely rescued the production of these signals in the *RSp0334* mutant (Fig. 1K).

I.130: and elsewhere: In the light of the identity of the identified 2',3' cyclic GMP, similar information for 3',5'- cyclic GMP is relevant.

Response: Good suggestion, we have added more details in text (Line 152-165).

I.178: What has been the rationale to select WspR and A1S_1695 for comparison and not any other GGDEF diguanylate cyclase.

Response: Because of the GGDEF domain (GGEEF) of WspR in *Pseudomonas aeruginosa* PAO1 and the GGDEF domain (GGEEF) of A1S_1695 in *Acinetobacter*

baumannii ATCC 17978 can efficiently catalyze the synthesis of bis-3',5'-c-di-GMP from GTP (Line 195-197).

The related references are shown:

1. Huangyutitham V, Güvener ZT, Harwood CS. Subcellular clustering of the phosphorylated WspR response regulator protein stimulates its diguanylate cyclase activity. *mBio*. 2013 May 7;4(3):e00242-13. PMID: 23653447; PMCID: PMC3663191.
2. Guo Q, Cui B, Wang M, Li X, Tan H, Song S, Zhou J, Zhang LH, Deng Y. Elongation factor P modulates *Acinetobacter baumannii* physiology and virulence as a cyclic dimeric guanosine monophosphate effector. *Proc Natl Acad Sci U S A*. 2022 Oct 11;119(41):e2209838119. PMID: 36191190; PMCID: PMC9564936.

I.229: binding domain

Response: Good suggestion, we have revised it as suggested.

I.323: Describe in more quantitative terms.

Response: Good suggestion, we have added more details in text (Line 341-343).

I.338: 3',5' cyclic GMP would be one of the closest controls.

Response: Good suggestion, we have supplemented this experiment and added the description in the text (Line 354-358).

Figure 2A: Highlight the relevant amino acids. Also, which part of the proteins is shown?

Response: Good suggestion, we have added more detail about the relevant amino acids in the figure legend.

Multiple sequence alignment of the GGDQF domain of RSp0334 in *R. solanacearum*, GGEEF domains of WspR in *P. aeruginosa* PAO1 and A1S_1695 in *A. baumannii* ATCC 17978 was performed with the ClustalW program and colored using GeneDoc 2.7. The key amino acid residues are marked in red and asterisk.

Figure 5A and Extended Data Fig. 6 E-J: Use the same maximum scale value on the y-axis for direct comparison.

Response: Good suggestion, we have revised it as suggested.

I. 599: Describe the principle of construction of reporter plasmids. Which part of the promoter regions have been used?

Response: Good suggestion, we have added more details as suggested (Line 449-452).

Extended Data Fig. 2: Please show the control spectra for comparison.

Response: Good suggestion, we have added the spectra of bis-3',5'-c-di-GMP as the control.

Extended Data Fig. 3: Peaks in C and D show the same retention time.

Response: Good suggestion, we have added more details in Supplementary Fig. 3H-I (RT=6.52 min) and the figure legend.

Extended Data Fig. 4: Explain the different colours used for the graphs.

Response: Good suggestion, we have added more details in the figure legend.

Extended Data Fig. 5D: Is the GGDQF domain a dimer?

Response: Good suggestion. Previous study found that the GGDEF domain of DGCs is emblematic of this control; since only one GTP molecule can bind in the active site, a dimerization step is mandatory for the cyclization reaction to occur and yield c-di-GMP. Consequently, multidomain DGCs usually regulate their activity by allowing the GGDEF domains to dimerize (active dimer) or forcing them apart (inactive conformations), functioning as conformational switches (Scheme 2) (Mantoni F, *et al.* Studying GGDEF domain in the act: minimize conformational frustration to prevent artefacts. Life (Basel). 2021 Jan 6;11(1):31. PMID: 33418960; PMCID: PMC7825114.). Similarly, the GGDQF domain of RSp0334 efficiently catalyzed the synthesis of 2',3'-c-di-GMP after mixing with 2',3'-cGMP for 10 min (Extended Data Fig. 5D), and we found the GGDQF domain of RSp0334 forms dimer, which was shown as follows (red box represents dimer).

Extended Data Fig. 10: mutated target gene promoter sequence or binding motif deleted? **Response:** The potential binding sites 5'-CCTGCCCGAC-3', 5'-GCAAATTCGG-3', 5'-CGAACCCGAC-3', and 5'-GGAAGTCGCC-3' were then deleted from the promoter regions of *phcB*, *soll*, *trpEG* and *epsA*, respectively (Line 291-293), and we have added more details in the figure legend.

Reviewer #4 (Remarks to the Author):

The authors have discovered a novel bacterial intracellular signaling mechanism that involves 2',3'-cyclic guanosine monophosphate (2',3'-cGMP), 2',3'-cyclic diguanosine monophosphate (2',3'-c-di-GMP) and two bacterial proteins, namely RSp0334 and RSp0980. Their results support the concept that in the plant pathogen *Ralstonia solanacearum* RSp0334 is an enzyme that converts 2',3'-cGMP to 2',3'-c-di-GMP, a reaction that lowers intracellular concentrations of 2',3'-cGMP while increasing intracellular concentrations of 2',3'-c-di-GMP. 2',3'-cGMP binds to its receptor RSp0980 and thereby prevents signaling by RSp0980. In the absence of 2',3'-cGMP, RSp0980 binds to the promoter region of a large number of genes and regulates cellular phenotype including virulence and quorum sensing signaling systems. Impairment of this signaling mechanism reduces the *R. solanacearum*-induced wilting of tomato plants. The authors provide evidence that this system exists also in *Salmonella typhimurium* (a human pathogen) and likely exists in a wide spectrum of bacteria. The RSp0334/2',3'-cGMP/RSp0980 pathway appears to be an important signaling mechanism in at least some species of bacteria. There are a few questions that should be addressed:

1. The K_m of RSp0334 for 2',3'-cGMP is approximately 100 μM and the K_d of 2',3'-cGMP for RSp0980 is approximately 10 μM . These data suggest that the intracellular concentrations of 2',3'-cGMP must be in the micromolar range to interact with these proteins. What is the intracellular level of 2',3'-cGMP in *Ralstonia solanacearum*? And how much do the intracellular concentrations of 2',3'-cGMP vary in bacteria? Are the levels within a reasonable range relative to the concentrations required to interact with the proposed components?

Response: We detected the intracellular concentrations of 2',3'-cGMP in *Ralstonia solanacearum* GMI1000 and the result showed that the intracellular concentrations of 2',3'-cGMP is $26.1 \pm 1.7 \mu\text{M}$ in *R. solanacearum* GMI1000 (Fig.S2K).

2. In the Discussion, the authors state that 3',5'-c-di-GMP is a potent immunostimulant in mammals that induces the production of type I interferons (IFNs) by directly binding to STING and therefore it would be worth investigating the potential role of 2',3'-c-di-GMP in the host (lines 378-381). InvivoGen sells 2',3'-c-di-GMP and the brochure provided by

InVivoGen describes 2',3'-c-di-GMP as a potent stimulator of type I IFNs (even more potent than 3',5'-c-di-GMP). Apparently, the question as to whether 2',3'-c-di-GMP interacts with STING has already been answered. The authors should discuss this.

Response: Good suggestion, we have added more details in the discussion part as suggested (Line 410-412).

A recent publication in bioRxiv showed that 2',3'-c-di-GMP acts as a novel potent agonist of STING in *Drosophila*, with functional relevance for the induction of antiviral immunity⁵¹. Therefore, it is also worth investigating the potential role of 2',3'-c-di-GMP in the mammal.

3. Recently, 2',3'-cGMP was discovered for the first time to exist in vivo and under physiological conditions in a mammalian species (American Journal of Physiology-Regulatory, Integrative and Comparative Physiology 316: R783–R790, 2019. PMC6620655, PMID: 30789788). In light of this information and the authors' current findings, the authors should discuss the possibility that 2',3'-cGMP may direct biochemical processes in mammals via a similar mechanism as described in bacteria.

Response: Good suggestion, we have added more details in the discussion as suggested (Line 415-422).

A previous study identified 2',3'-cGMP in plants and found that TIR proteins exhibit 2',3'-cAMP/cGMP synthetase activity *in vitro*⁴³. In addition, mRNA turnover promoted by RNase I can produce 2',3'-cNMPs in *Escherichia coli* and *S. typhimurium*⁴². However, we have not found protein homologs of TIR and RNase I in *R. solanacearum*, suggesting that there may be a unique synthetase for this signal in *R. solanacearum*. In addition, whether there are the similar 2',3'-cGMP to 2',3'-c-di-GMP pathways in eukaryotes is also worth exploring. Given that PDEs containing an HD, DHH or calcineurin-like PDE domain could hydrolyze 2',3'-cGMP to 2'-GMP and 3'-GMP⁵², our research may reveal a new conversion pathway for 2',3'-cGMP.

4. Related to item #3, the AJP article mentioned above also shows that in vivo 2',3'-cyclic nucleotide 3'-phosphodiesterase (CNPase) metabolizes 2',3'-cGMP to 2'-GMP. Is there evidence for the existence of a homolog of the mammalian CNPase in bacteria? This could be a key player regulating 2',3'-cGMP levels.

Response: Good suggestion, however, we did not find the homolog of mammalian CNPase in *Ralstonia solanacearum* GMI1000. And we have listed all the homologs of the synthases and degrading enzymes of 2',3'-cNMP that have been identified so far, and we could not find any homolog of 2',3'-cGMP synthase in *Ralstonia solanacearum* GMI1000, suggesting that there might be a distinctive 2',3'-cGMP synthase in *R. solanacearum*, which is worth to be investigated in the future.

Table. Potential homologs of 2',3'-cNMP synthetase and hydrolase in *R. solanacearum*

Synthetase	Accession	Species	Reference	Identify
NLR immune receptors	7VU8_B	Nicotiana benthamiana	1	NA
RNase I	AAC73712	Escherichia coli	2	NA

Hydrolase	Accession	Species	Reference	Identify
tRNA-NT	P06961	Escherichia coli	3	53.68%
SpdA	WP_100669706	Sinorhizobium meliloti	4	33.33%
NUDT7	OAP01189	Arabidopsis thaliana	1	33.33%
PdeA	WP_201421553	Myxococcus xanthus	5	28.25%
PdeB	NOJ79288	Myxococcus xanthus	5	28%
YtqI	AAC00337	Bacillus subtilis	6	NA
CNPase	AH004086.2	Homo sapiens	7	NA

NA, not applicable

(1) Yu, D., et al., TIR domains of plant immune receptors are 2',3'-cAMP/cGMP synthetases mediating cell death. *Cell* **2022**, 185 (13), 2370-2386.

(2) Fontaine, B. M., et al., RNase I regulates *Escherichia coli* 2',3'-cyclic nucleotide monophosphate levels and biofilm formation. *Biochem. J.* **2018**, 475 (8), 1491-1506.

(3) Yakunin, A. F., et al., The HD domain of the *Escherichia coli* tRNA nucleotidyltransferase has 2',3'-cyclic phosphodiesterase, 2'-nucleotidase, and phosphatase activities. *J. Biol. Chem.* **2004**, 279 (35), 36819-36827.

(4) Mathieu-Demaziere, C.; Poinso, V.; Masson-Boivin, C.; Garnerone, A. M.; Batut, J., Biochemical and functional characterization of SpdA, a 2', 3'-cyclic nucleotide phosphodiesterase from *Sinorhizobium meliloti*. *BMC Microbiol.* **2013**, 13, 268.

(5) Kimura, Y.; Okazaki, N.; Takegawa, K., Enzymatic characteristics of two novel *Myxococcus xanthus* enzymes, PdeA and PdeB, displaying 3',5'- and 2',3'-cAMP phosphodiesterase, and phosphatase activities. *FEBS Lett* **2009**, 583 (2), 443-8.

(6) Rao, F.; Qi, Y.; Murugan, E.; Pasunooti, S.; Ji, Q., 2',3'-cAMP hydrolysis by metal-dependent phosphodiesterases containing DHH, EAL, and HD domains is non-specific: Implications for PDE screening. *Biochem. Biophys. Res. Commun.* **2010**, 398 (3), 500-505.

(7) Thompson RJ. 2',3'-cyclic nucleotide-3'-phosphohydrolase and signal transduction in central nervous system myelin. *Biochem Soc Trans.* **1992** Aug;20(3):621-6.

5. Does *Ralstonia solanacearum* secrete 2',3'-cGMP or 3',5'-c-di-GMP into the culture medium? If so, this could provide a pathway for extracellular signaling. This should be discussed.

Response: Good suggestion, we detected the concentration of 2',3'-cGMP in the culture medium is 0.142 ± 0.047 nM.

6. In the text and figures, the authors mention 2'-cGMP and 3'-cGMP. Don't you mean 2'-GMP and 3'-GMP? Not sure what the structure of 2'-cGMP and 3'-cGMP would be.

Response: Thank you very much for your suggestion, we have revised it (Line 250, Line 421, Line 434 and Supplementary Fig. 8A-8B).

REVIEWER COMMENTS

Reviewer #1 (Remarks to the Author):

This revised manuscript by Li et. al. has been significantly improved. I am still skeptical of the synthesis of 2'3'-c-di-GMP by RSp0334, and do not find the author's explanation for the mechanism to be very convincing. Indeed, I do not believe they demonstrate production of this molecule *in vivo*, and maybe it is just an artifact of the *in vitro* synthesis experiments. However, given that their argument for the biological function of RSp0334 is a reduction of the substrate, 2'3'-cGMP, and their data support synthesis of 2'3'-c-di-GMP *in vitro*, then I think having this research published will be a stepping stone to determining if the 2'3'-c-di-GMP synthesis occurs *in vivo* and has biological function. I would like the authors to propose the enzymatic mechanism that they gave the reviewers in the discussion of the paper as I am certain this will be a major question for this paper by anyone in the field.

Reviewer #2 (Remarks to the Author):

Reviewer #2 (Remarks to the Author):

I was very surprised that the authors claim that they identified a “new bacterial intracellular signal, 2',3'-cyclic guanosine monophosphate (2',3'-cGMP)”. In fact, 2',3'-cNMP has previously been proposed as a novel class of bacterial signals by the Weinert lab, see e.g.

Fontaine B.M., Martin K.S., Garcia-Rodriguez J.M., Jung C., Briggs L., Southwell J.E., Jia X., Weinert E.E. RNase I regulates *Escherichia coli* 2',3'-cyclic nucleotide monophosphate levels and biofilm formation. *Biochem. J.* 2018;475:1491–1506. doi: 10.1042/BCJ20170906.

Duggal Y., Kurasz J.E., Fontaine B.M., Marotta N.J., Chauhan S.S., Karls A.C., Weinert E.E. Cellular effects of 2',3'-cyclic nucleotide monophosphates in Gram-negative bacteria. *J. Bacteriol.* 2022;204:e0020821. doi: 10.1128/JB.00208-21.

Fontaine B.M., Duggal Y., Weinert E.E. 2',3'-Cyclic mononucleotide metabolism and possible roles in bacterial physiology. In: Chou S.H., Guiliani N., Lee V.T., Römling U., editors. *Microbial Cyclic Di-Nucleotide Signaling*. Springer Nature; Cham, Switzerland: 2020. pp. 627–637.

Chauhan S.S., Marotta N.J., Karls A.C., Weinert E.E. Binding of 2',3'-cyclic nucleotide monophosphates to bacterial ribosomes inhibits translation. ACS Cent. Sci. 2022;8:1518–1526. doi: 10.1021/acscentsci.2c00681.

Of these publications only the Duggal et al. paper is mentioned in the manuscript and this only in the context of 2',3'-cNMPs production, while this paper provides clear evidence that 2',3'-cNMPs signaling is widespread in bacteria (and eukaryotes), where it affects global gene expression, flagellar motility, biofilm formation and acid tolerance. Hence, not only the novelty of the data presented in this paper is limited it also fails to discuss the results in the light of the current state of knowledge.

It has been reported that in *E. coli* and *S. Typhimurium* 2',3'-cNMPs are generated through hydrolysis of RNA by RNAses. The important question of how this signal is produced in *Ralstonia* is not addressed in the present study. It rather identified an enzyme, RSp0334, which converts 2',3'-cGMP to 2',3'-c-di-GMP and thus inactivates the signal. However, the effect in vivo is not impressive, as inactivation of RSp0334 increased the cellular 2',3'-cGMP by about 50%. Notably, enzymes that hydrolyze 2',3'-cGMP such as 2',3' cyclic phosphodiesterases have been extensively characterized in the enterobacteriaceae and have also been identified in eukaryotes. Overexpression of such enzymes was shown to fully deplete the cellular 2',3'-cNMP levels and were employed to investigate the functions of this signal in gene regulation. Are homologous enzymes present in *Ralstonia*? Do they play role in modulating 2',3'-cNMP levels? Why was the approach to enzymatically deplete 2',3'-cGMP not used to investigate the role of this signal in *Ralstonia*?

Response: Thanks a lot for your valuable comments and suggestions.

Firstly, in general, the novelties in our study are:

- (1). We identified that the transcriptional regulator, RSp0980, is a novel receptor of 2',3'-cGMP; and 2',3'-cGMP specifically binds to RSp0980 with high affinity and thus abolishes the interaction between RSp0980 and the promoters of target genes.
- (2). The LLARLGGDQF region of RSp0334 catalyzes 2',3'-cGMP to (2',5')(3',5')-cyclic diguanosine monophosphate (2',3'-c-di-GMP), and controls the important phenotypes through decreasing the intracellular 2',3'-cGMP levels.
- (3). The LLD amino acid sites of RSp0334 determines the enzyme characteristics of RSp0334.
- (4). The RSp0334/2',3'-cGMP/RSp0980 signaling system also exists in the human pathogen *Salmonella typhimurium*, and may exist in other bacteria.

We have added these papers mentioned above as references and discuss them in the discussion part in Line 368-371, Line 415-419

> While I appreciate that the missing references were included, I still feel that they are not appropriately discussed in the light how 2',3'-cGMP is synthesized and perceived in other bacteria. The differences and communalities should be highlighted to show where in fact the novelty is.

> The authors claim that the RSp0334/2',3'-cGMP/RSp0980 signaling system also exists in the human pathogen *Salmonella typhimurium*, and may exist in other bacteria. To support this argument the authors present homologs of RSp0334 and RSp0980 in Table S8. However, a closer look at this table shows that of the 11 homologous proteins only those of *Salmonella* and *Ruegeria mobilis* possess the LLARLGGDEF domain while the remaining ones not even have the minimal L542, L545 and D548 motive required for function and thus cannot have a function in 2',3'-cGMP signaling.

It is also not clear how the authors identified homologous proteins. For example, in *P. aeruginosa* the protein with the highest homology to RSp0334 is PA0285, which has a LLSRLGGDEF domain, and not WP_003118137.1 as stated in the table. Notably, several other homologs in PAO1 also have the L542, L545 and D548 motive, some of which are well-characterized enzymes affecting the cellular c-di-GMP level although, according to the definitions of the authors, should be involved in 2',3'-cGMP signaling. Yet, there is no evidence that this would be the case.

Secondly, we just detected the ratio of 2',3'-cGMP in the RSp0334 deletion mutant strain to the wild-type strain at a time point, and we believe that this ratios between the two strains are dynamically changed at different time points.

>This could well be (as suggested previously for *E. coli* and *Salmonella*), but in this case it would have been interesting to know under which condition the deletion of RSp0334 would show a more convincing effect on the cellular 2',3'-cGMP level.

Thirdly, we have listed all the homologs of the synthases and degrading enzymes of 2',3'-cNMP that have been identified so far, and we could not find any homolog of 2',3'-cGMP synthase in *Ralstonia solanacearum* GMI1000, suggesting that there might be a distinctive 2',3'-cGMP synthase in *R. solanacearum*, which is worth to be investigated in the future.

> I still believe that without the identification of the 2',3'-cGMP biosynthesis pathway this story will not be complete.

> The authors did not respond to my suggestion that 2',3'-cGMP levels could be depleted by the use of 2',3' cyclic phosphodiesterases to investigate the role of this signal in *Ralstonia*.

Table. Potential homologs of 2',3'-cNMP synthetase and hydrolase in *R. solanacearum*

NA, not applicable

- (1) Yu, D., et al., TIR domains of plant immune receptors are 2',3'-cAMP/cGMP synthetases mediating cell death. *Cell* 2022, 185 (13), 2370-2386.
- (2) Fontaine, B. M., et al., RNase I regulates *Escherichia coli* 2',3'-cyclic nucleotide monophosphate levels and biofilm formation. *Biochem. J.* 2018, 475 (8), 1491-1506.
- (3) Yakunin, A. F., et al., The HD domain of the *Escherichia coli* tRNA nucleotidyltransferase has 2',3'-cyclic phosphodiesterase, 2'-nucleotidase, and phosphatase activities. *J. Biol. Chem.* 2004, 279 (35), 36819-36827.
- (4) Mathieu-Demaziere, C.; Poinso, V.; Masson-Boivin, C.; Garnerone, A. M.; Batut, J., Biochemical and functional characterization of SpdA, a 2', 3' cyclic nucleotide phosphodiesterase from *Sinorhizobium meliloti*. *BMC Microbiol.* 2013, 13, 268.
- (5) Kimura, Y.; Okazaki, N.; Takegawa, K., Enzymatic characteristics of two novel *Myxococcus xanthus* enzymes, PdeA and PdeB, displaying 3',5'- and 2',3'-cAMP phosphodiesterase, and phosphatase activities. *FEBS Lett* 2009, 583 (2), 443-8.
- (6) Rao, F.; Qi, Y.; Murugan, E.; Pasunooti, S.; Ji, Q., 2',3'-cAMP hydrolysis by metal-dependent phosphodiesterases containing DHH, EAL, and HD domains is non-specific: Implications for PDE screening. *Biochem. Biophys. Res. Commun.* 2010, 398 (3), 500-505.
- (7) Thompson RJ. 2',3'-cyclic nucleotide-3'-phosphohydrolase and signal transduction in central nervous system myelin. *Biochem Soc Trans.* 1992 Aug;20(3):621-6.

The paper then switches to 2',3'-cGMP perception and identified RSp0980 as a potential receptor for the molecule. However, the EMSA assays shown in Fig. 4 are unconvincing, the effect of 2',3'-cGMP is at the best very weak, the concentrations used appear very high. The proposed consensus sequence for RSp0980 binding is feeble. Moreover, RSp0980 is annotated as probable nitrate/nitrite response regulator. This predicted function of RSp0980 is not only supported by the fact the genes encoding the pathway for dissimilatory nitrate reduction is located next to the gene but also by the fact that homologs of RSp0980 have been demonstrated to be involved in nitrate sensing and expression of the dissimilatory nitrate reductase in other bacteria (see e.g. Mangalea MR, Borlee BR. *Sci Rep.* 2022 Jan 7;12(1):203. doi: 10.1038/s41598-021-04053-6. and Li W et al. *Int J Mol Sci.* 2022 Jun 29;23(13):7220. doi: 10.3390/ijms23137220). Finally, a recent study in *E. coli* showed that 2',3'-cNMPs bind to bacterial ribosomes, inhibit translation in vitro, and modulate growth rates, suggesting a potential mechanism for rapidly altering translation (Chauhan SS, Marotta NJ, Karls AC, Weinert EE. *ACS Cent Sci.* 2022 Nov 23;8(11):1518-1526. doi: 10.1021/acscentsci.2c00681). Such a mechanism that has a global effect on cellular physiology may also explain the phenotypes of the mutants reported in this study.

Response: Thanks a lot for your valuable comments and suggestions.

Firstly, RSp0980 bound strongly to 2',3'-cGMP, with an estimated dissociation constant (KD) of $7.14 \pm 0.64 \mu\text{M}$ (Line 245), 2',3'-cGMP also significantly inhibited RSp0980 from binding to the promoter of the target gene at $10 \mu\text{M}$ (Fig. 4).

> My concern remains that the inhibition of binding of 0980 to the target promoters in the presence 2',3'-cGMP of is unconvincing.

Secondly, we examined the effect of nitrate on the phenotypes of *R. solanacearum* and we found that the exogenous addition of NaNO₃ at low concentration didn't affect the biofilm formation and motility of the wild-type strain, but only addition of high concentrations of NaNO₃ at 1mM and 10 mM obviously inhibited the biofilm formation and motility of *R. solanacearum* wild-type strain. Interestingly, exogenous addition of NaNO₃ did not affect the phenotypes of the RSp0980 deletion mutant, suggesting that RSp0980 is possible to be involved in nitrate sensing, we have added the discussion and relevant references in Line 399-401.

> I appreciate that the effect of nitrate on biofilm formation and motility has been investigated. They are in line with the results reported for *B. pseudomallei* in that nitrate inhibits both biofilm formation and motility and that inactivation of narL, the homolog of RSp0980, is no longer responsive to the presence of nitrate. In *B. pseudomallei*, the NarX/NarL two component system was shown to control of expression adjacent genes encoding enzymes for nitrate-nitrite respiration along with several other functions affecting biofilm formation and secondary metabolite production. As in *B. pseudomallei*, the homologous genes for nitrate respiration are next to RSp0980. It would therefore be important to elucidate its role for nitrate-nitrite respiration, as this appears to be the primary function of RSp0980. In any case, while the authors suggest in the revised version of the manuscript that RSp0980 is possibly involved in nitrate sensing they fail in providing a convincing model in how this would go together with the proposed sensing of 2',3'-cGMP by the same response regulator.

Finally, previous study showed that 2',3'-cNMPs alter bacterial growth by bind to bacterial ribosomes ((Chauhan SS, Marotta NJ, Karls AC, Weinert EE. ACS Cent Sci. 2022 Nov 23;8(11):1518-1526). However, the deletion of RSp0334 had little effect on the growth of bacterial cells in different media (Supplementary Fig. 1C-E) but resulted in significant defects in phenotypes including motility, biofilm formation, cellulase production, and extracellular polysaccharide (EPS) production, and in trans expression of RSp0334 restored all of the phenotypes of the RSp0334 deletion mutant to those of the wild-type strain (Fig. 1C-F) (Line107-111), suggesting that 2',3'-cGMP utilized a different mechanism in *R. solanacearum* from that in *E. coli*.

> In fact, the effect on growth reported by Chauhan et al. was not dramatic and is probably comparable with the differences seen between the *Ralstonia* wt and the RSp0334 deletion mutant 0.1 in TTC medium. Moreover, Chauhan et al. showed that a 2-fold increase in the 2',3'-cGMP level does not affect growth. As the effect of deleting RSp0334 in *Ralstonia* increases the 2',3'-cGMP level by less than 2-fold, no effect on growth can be expected and thus an inhibition of translation cannot be excluded.

Reviewer #3 (Remarks to the Author):

This is the revised version of a manuscript previously submitted to Nature Communications. The authors have to a great extent satisfactorily answered to my comments. I have the following follow up comments.

The major novel finding is perhaps not the role of 2',3'-cyclic GMP as a second messenger, but the unconventional functionality of RSp0334 in catalyzing 2',3' cyclic GMP to 2'3' cyclic di-GMP. This should be reflected in the abstract.

The phosphorylation site for the NarL homologue RSp0980 on Thr at position 12 is unconventional. Can the authors comment on this in the manuscript.

PdeR of *Salmonella typhimurium* contains a GGDEF motif instead of a GGDQF motif. What is the effect of the E>Q substitution? Does PdeR make both 2'3' cyclic di-GMP and 3'3' cyclic di-GMP? Genetic experiments performed by Ahmad et al, BMC Microbiol, 2017 suggest conventional catalytic cyclase and phosphodiesterase activities by PdeR.

Reviewer #4 (Remarks to the Author):

The authors have carefully, methodically and convincingly addressed all comments by all 4 Reviewers. The authors have also revised the manuscript according to the suggestions and concerns of all Reviewers, including addition of new data.

Reviewer #1 (Remarks to the Author):

This revised manuscript by Li et. al. has been significantly improved. I am still skeptical of the synthesis of 2',3'-c-di-GMP by RSp0334, and do not find the author's explanation for the mechanism to be very convincing. Indeed, I do not believe they demonstrate production of this molecule *in vivo*, and maybe it is just an artifact of the *in vitro* synthesis experiments. However, given that their argument for the biological function of RSp0334 is a reduction of the substrate, 2',3'-cGMP, and their data support synthesis of 2',3'-c-di-GMP *in vitro*, then I think having this research published will be a stepping stone to determining if the 2',3'-c-di-GMP synthesis occurs *in vivo* and has biological function. I would like the authors to propose the enzymatic mechanism that they gave the reviewers in the discussion of the paper as I am certain this will be a major question for this paper by anyone in the field.

Response: Thanks a lot for your good suggestion. To further confirm the role of RSp0334 in the biosynthesis of 2',3'-c-di-GMP in *R. solanacearum*. Firstly, we continued to detect the intracellular levels of 2',3'-c-di-GMP in the wild-type, *RSp0334* deletion mutant and complemented strains, and found that the intracellular level of 2',3'-c-di-GMP was significantly decreased in the *RSp0334* deletion mutant strain compared to that in the wild-type strain (**The response Fig. 1**), we also add this new data in **Supplementary Fig. 3J** in this revised version of our manuscript. Secondly, we also overexpressed RSp0334 in *E. coli* and then detected the intracellular levels of 2',3'-cGMP and 2',3'-c-di-GMP. The results showed that the intracellular level of 2',3'-cGMP was significantly decreased while the intracellular level of 2',3'-c-di-GMP was significantly increased in the *E. coli* (*RSp0334*) strain compared to those in the *E. coli* wild-type strain (**The response Fig. 2**). Together, both the *in vivo* and *in vitro* data suggest that Rsp0334 catalyzes 2',3'-cGMP to 2',3'-c-di-GMP.

For the biological function of 2',3'-c-di-GMP, there is a recent publication in *Immunity* showed that 2',3'-c-di-GMP acts as a novel potent agonist of STING in *Drosophila* (Line 412- Line 413), with functional relevance for the induction of antiviral immunity (Cai, H. *et al.* The virus-induced cyclic dinucleotide 2',3'-c-di-GMP mediates STING-dependent antiviral immunity in *Drosophila*. *Immunity*, 2023 Aug 28:S1074-7613(23)00362-X. doi: 10.1016/j.immuni.2023.08.006). We suppose that 2',3'-c-di-GMP is a potential signal in *R. solanacearum*, and we will continue to study the signaling pathway of 2',3'-c-di-GMP in *R. solanacearum* in our next project. Finally, we have also added the discussion of the enzymatic mechanism of Rsp0334 as suggested (Line 384-Line 385).

The response Fig. 1. Influence of RSp0334 in the intracellular level of 2',3'-c-di-GMP in *R. solanacearum*

The response Fig. 2. Influence of *in trans* expression of RSp0334 in the intracellular levels of 2',3'-cGMP and 2',3'-c-di-GMP in *E. coli*

Reviewer #2 (Remarks to the Author):

I was very surprised that the authors claim that they identified a “new bacterial intracellular signal, 2',3'-cyclic guanosine monophosphate (2',3'-cGMP)”. In fact, 2',3'-cNMP has previously been proposed as a novel class of bacterial signals by the Weinert lab, see e.g.

Fontaine B.M., Martin K.S., Garcia-Rodriguez J.M., Jung C., Briggs L., Southwell J.E., Jia X., Weinert E.E. RNase I regulates *Escherichia coli* 2',3'-cyclic nucleotide monophosphate levels and biofilm formation. *Biochem. J.* 2018;475:1491–1506. doi: 10.1042/BCJ20170906.

Duggal Y., Kurasz J.E., Fontaine B.M., Marotta N.J., Chauhan S.S., Karls A.C., Weinert E.E. Cellular effects of 2',3'-cyclic nucleotide monophosphates in Gram-negative bacteria. *J. Bacteriol.* 2022;204:e0020821. doi: 10.1128/JB.00208-21.

Fontaine B.M., Duggal Y., Weinert E.E. 2',3'-Cyclic mononucleotide metabolism and possible roles in bacterial physiology. In: Chou S.H., Guiliani N., Lee V.T., Römling U., editors. *Microbial Cyclic Di-Nucleotide Signaling*. Springer Nature; Cham, Switzerland: 2020. pp. 627–637.

Chauhan S.S., Marotta N.J., Karls A.C., Weinert E.E. Binding of 2',3'-cyclic nucleotide monophosphates to bacterial ribosomes inhibits translation. *ACS Cent. Sci.* 2022;8:1518–1526. doi: 10.1021/acscentsci.2c00681.

Of these publications only the Duggal et al. paper is mentioned in the manuscript and this only in the context of 2',3'-cNMPs production, while this paper provides clear evidence that 2',3'-cNMPs signaling is widespread in bacteria (and eukaryotes), where it affects global gene expression, flagellar motility, biofilm formation and acid tolerance. Hence, not only the novelty of the data presented in this paper is limited it also fails to discuss the results in the light of the current state of knowledge.

It has been reported that in *E. coli* and *S. Typhimurium* 2',3'-cNMPs are generated through hydrolysis of RNA by RNases. The important question of how this signal is produced in *Ralstonia* is not addressed in the present study. It rather identified an enzyme, RSp0334, which

converts 2',3'-cGMP to 2',3'-c-di-GMP and thus inactivates the signal. However, the effect *in vivo* is not impressive, as inactivation of RSp0334 increased the cellular 2',3'-cGMP by about 50%. Notably, enzymes that hydrolyze 2',3'-cGMP such as 2',3' cyclic phosphodiesterases have been extensively characterized in the enterobacteriaceae and have also been identified in eukaryotes. Overexpression of such enzymes was shown to fully deplete the cellular 2',3'-cNMP levels and were employed to investigate the functions of this signal in gene regulation. Are homologous enzymes present in *Ralstonia*? Do they play role in modulating 2',3'-cNMP levels? Why was the approach to enzymatically deplete 2',3'-cGMP not used to investigate the role of this signal in *Ralstonia*?

Response: Thanks a lot for your valuable comments and suggestions.

Firstly, in general, the novelties in our study are:

- (1). We identified that the transcriptional regulator, RSp0980, is a novel receptor of 2',3'-cGMP; and 2',3'-cGMP specifically binds to RSp0980 with high affinity and thus abolishes the interaction between RSp0980 and the promoters of target genes.
- (2). The LLARLGGDQF region of RSp0334 catalyzes 2',3'-cGMP to (2',5')(3',5')-cyclic diguanosine monophosphate (2',3'-c-di-GMP), and controls the important phenotypes through decreasing the intracellular 2',3'-cGMP levels.
- (3). The LLD amino acid sites of RSp0334 determines the enzyme characteristics of RSp0334.
- (4). The RSp0334/2',3'-cGMP/RSp0980 signaling system also exists in the human pathogen *Salmonella typhimurium*, and may exist in other bacteria.

We have added these papers mentioned above as references and discuss them in the discussion part in Line 368-371, Line 415-419

> While I appreciate that the missing references were included, I still feel that they are not appropriately discussed in the light how 2',3'-cGMP is synthesized and perceived in other bacteria. The differences and communalities should be highlighted to show where in fact the novelty is.

Response: We thanks a lot for the great work of Weinert lab in the study of 2',3'-cNMPs in *E. coli* and *S. typhimurium*. And their research results also give us a lot of useful information for our study of 2',3'-cGMP in *R. solanacearum*. However, we also discover some interesting findings of 2',3'-cGMP in *R. solanacearum*, which are different from those in *E. coli*, suggesting the distinct regulatory mechanisms of 2',3'-cGMP signaling system in different bacterial species.

Firstly, the published studies showed that 2',3'-cNMPs are produced by RNase I in *E. coli* and *S. typhimurium*, and 2',3'-cGMP is produced by TIR protein in plants. We have not found protein homologs of TIR and RNase I of both *E. coli* and *S. typhimurium* in *R. solanacearum*, but we have found a homolog of ribonuclease T1 from *Cupriavidus plantarum* in *R. solanacearum* GMI1000. The *in vitro* enzyme experiment showed that the homologous protein RSc2766 catalyzes mRNA to 2',3'-cGMP (**The response Fig. 3**), we add this new data in **Supplementary Fig. 13**. We will also continue to study the role of RSc2766 in *R. solanacearum* in the future.

Secondly, our study revealed that the GGDQF domain of RSp0334 can convert 2',3'-cGMP to 2',3'-c-di-GMP in the alkaline condition of pH 8.0 (**Supplementary Fig. 5**) (Line 384-

Line 385). In addition, as your suggestion, we have tested a HD-GYP protein, RSc0592, and found that it could degrade 2',3'-cGMP, suggesting that the RSc0592 is a new 2',3' cyclic phosphodiesterase in *R. solanacearum* (The response Fig. 4). Interestingly, we also found that overexpression of RSc0592 could also rescue the impaired phenotypes of the *Rsp0334* deletion mutant (The response Fig. 5).

Thirdly, Chauhan, S.S. *et al.* found that 2',3'-cNMP can bind to bacterial ribosomes and then result in decreased bacterial growth. In this study, we found that 2',3'-cGMP was shown to bind to the REC domain of RSp0980 with high affinity and abolish the binding of RSp0980 to target promoter DNA (Fig. 3N-P and Fig. 4I-L), suggesting that 2',3'-cGMP might employ a different signaling pathways in *R. solanacearum*.

Finally, as your good suggestion, we have measured the time course of 2',3'-cGMP production at various growth stages in *R. solanacearum*. The results showed that 2',3'-cGMP concentrations increased and peaked at OD₆₀₀=1.5, followed by a decline in the 2',3'-cGMP concentration (Supplementary Fig. S2L, The response Fig. 6).

The response Fig. 3. The *in vitro* biosynthesis of 2',3'-cGMP by RSc2766 (Supplementary Fig. 13)

The response Fig. 4. Degradation of 2',3'-cGMP by the HD-GYP protein RSc0592 *in vitro*

The response Fig. 5. *In trans* expression of RSc0592 rescues the impaired phenotypes of the deletion mutant of *Rsp0334*

The response Fig. 6. The time course of 2',3'-cGMP production in *R. solanacearum* (Supplementary Fig. 2L)

> The authors claim that the RSp0334/2',3'-cGMP/RSp0980 signaling system also exists in the human pathogen *Salmonella typhimurium*, and may exist in other bacteria. To support this argument the authors present homologs of RSp0334 and RSp0980 in Table S8. However, a closer look at this table shows that of the 11 homologous proteins only those of *Salmonella* and *Ruegeria mobilis* possess the LLARLGGDEF domain while the remaining ones not even have the minimal L542, L545 and D548 motive required for function and thus cannot have a function in 2',3'-cGMP signaling.

It is also not clear how the authors identified homologous proteins. For example, in *P. aeruginosa* the protein with the highest homology to RSp0334 is PA0285, which has a LLSRLGGDEF domain, and not WP_003118137.1 as stated in the table. Notably, several other homologs in PA01 also have the L542, L545 and D548 motive, some of which are well-characterized enzymes affecting the cellular c-di-GMP level although, according to the definitions of the authors, should be involved in 2',3'-cGMP signaling. Yet, there is no evidence that this would be the case.

Response: Thanks for your good suggestion. We have listed the Table S8 for the convenience of the study of the potential homologs of RSp0334 and Rsp0980 in other bacterial species. We have deleted this table in this revised version of our manuscript as some of the proteins are not the true homologs of Rsp0334.

Secondly, we just detected the ratio of 2',3'-cGMP in the RSp0334 deletion mutant strain to the wild-type strain at a time point, and we believe that this ratios between the two strains are dynamically changed at different time points.

>This could well be (as suggested previously for *E. coli* and *Salmonella*), but in this case it would have been interesting to know under which condition the deletion of RSp0334 would show a more convincing effect on the cellular 2',3'-cGMP level.

Response: Thanks for your good suggestion, we have detected the intracellular concentrations of 2',3'-cGMP in the wild-type, RSp0334 deletion mutant and complemented strains at different growth phases. The results showed that 2',3'-cGMP concentrations increased and peaked at OD₆₀₀=1.5, followed by decline in the 2',3'-cGMP concentration (**Supplementary Fig. 2L, The response Fig. 6**).

Thirdly, we have listed all the homologs of the synthases and degrading enzymes of 2',3'-cNMP that have been identified so far, and we could not find any homolog of 2',3'-cGMP synthase in *Ralstonia solanacearum* GMI1000, suggesting that there might be a distinctive 2',3'-cGMP synthase in *R. solanacearum*, which is worth to be investigated in the future.

> I still believe that without the identification of the 2',3'-cGMP biosynthesis pathway this story will not be complete.

> The authors did not respond to my suggestion that 2',3'-cGMP levels could be depleted by the use of 2',3' cyclic phosphodiesterases to investigate the role of this signal in *Ralstonia*.

Response: Thanks a lot for your two good suggestions. We have provided the detailed information of the biosynthesis and degradation of 2',3'-cGMP in the abovementioned response (**in the response Fig. 3, 4, 5**).

The paper then switches to 2',3'-cGMP perception and identified RSp0980 as a potential receptor for the molecule. However, the EMSA assays shown in Fig. 4 are unconvincing, the effect of 2',3'-cGMP is at the best very weak, the concentrations used appear very high. The proposed consensus sequence for RSp0980 binding is feeble. Moreover, RSp0980 is annotated as probable nitrate/nitrite response regulator. This predicted function of RSp0980 is not only supported by the fact the genes encoding the pathway for dissimilatory nitrate reduction is located next to the gene but also by the fact that homologs of RSp0980 have been demonstrated to be involved in nitrate sensing and expression of the dissimilatory nitrate reductase in other bacteria (see e.g. Mangalea MR, Borlee BR. *Sci Rep.* 2022 Jan 7;12(1):203. doi: 10.1038/s41598-021-04053-6. and Li W et al. *Int J Mol Sci.* 2022 Jun 29;23(13):7220. doi: 10.3390/ijms23137220). Finally, a recent study in *E. coli* showed that 2',3'-cNMPs bind to bacterial ribosomes, inhibit translation in vitro, and modulate growth rates, suggesting a potential mechanism for rapidly altering translation (Chauhan SS, Marotta NJ, Karls AC, Weinert EE. *ACS Cent Sci.* 2022 Nov 23;8(11):1518-1526. doi: 10.1021/acscentsci.2c00681). Such a mechanism that has a global effect on cellular physiology may also explain the phenotypes of the mutants reported in this study.

Response: Thanks a lot for your valuable comments and suggestions.

Firstly, RSp0980 bound strongly to 2',3'-cGMP, with an estimated dissociation constant (KD) of $7.14 \pm 0.64 \mu\text{M}$ (Line 245), 2',3'-cGMP also significantly inhibited RSp0980 from binding to the promoter of the target gene at $10 \mu\text{M}$ (Fig. 4).

> My concern remains that the inhibition of binding of 0980 to the target promoters in the presence 2',3'-cGMP of is unconvincing.

Response: Thanks a lot for your good comments. To answer this question, we firstly further confirm the binding between RSp0980 and 2',3'-cGMP by using two different methods. We used both microscale thermophoresis (MST) and Isothermal Titration Calorimetry (ITC) analysis and the results showed that RSp0980 tightly bound to 2',3'-cGMP with an estimated dissociation constant (KD) of $7.14 \pm 0.64 \mu\text{M}$ and $6.39 \pm 0.907 \mu\text{M}$, respectively (**Fig. 3L and 3N, The response Fig. 7**).

Secondly, we examined the effects of 2',3'-cGMP on the binding of RSp0980 to the promoters of *phcB*, *soll*, *trpEG* and *epsA* by EMSA assays with several more times. As shown in **Fig. 4I-L** in this revised version, the binding of RSp0980 to the target gene promoter probes was obviously inhibited when 2',3'-cGMP was present at $10 \mu\text{M}$ in the reaction mixtures (The molar concentration of 2',3'-cGMP: Rsp0980=1:1) (**The response Fig. 8**).

Thirdly, other cyclic nucleotide compounds were tested, and we found that RSp0980 did not bind to 2'-GMP, 3'-GMP, 3',5'-cAMP, or 3',5'-cGMP but bound weakly to 2',3'-cAMP and bis-3',5'-c-di-GMP, with estimated K_D values of $81.95 \pm 0.93 \mu\text{M}$ and $130.37 \pm 1.3 \mu\text{M}$, respectively (Supplementary Fig. 8A-F, **The response Fig. 9**). Consistent with the above results, the addition of exogenous 2',3'-c-di-GMP or bis-3',5'-c-di-GMP did not affect the binding of RSp0980 to the *epsA* promoter probe, while 2',3'-cAMP slightly affected the binding of RSp0980 to the *epsA* promoter probe at $100 \mu\text{M}$, suggesting that RSp0980 is a specific effector of 2',3'-cGMP (Supplementary Fig. 8G-I, **The response Fig. 9**).

The response Fig. 7. MST and ITC analysis of the binding of RSp0980 to 2',3'-cGMP (Fig. 3L and 3N)

The response Fig. 8. Effects of 2',3'-cGMP on the binding of RSp0980 to the target gene promoters (Fig. 4I-L)

The response Fig. 9. Analysis of the binding of RSp0980 to nucleotide molecules (Supplementary Fig. 8)

Secondly, we examined the effect of nitrate on the phenotypes of *R. solanacearum* and we found that the exogenous addition of NaNO_3 at low concentration didn't affect the biofilm formation and motility of the wild-type strain, but only addition of high concentrations of NaNO_3 at 1mM and 10 mM obviously inhibited the biofilm formation and motility of *R. solanacearum* wild-type strain. Interestingly, exogenous addition of NaNO_3 did not affect the phenotypes of the RSp0980 deletion mutant, suggesting that RSp0980 is possible to be involved in nitrate sensing, we have added the discussion and relevant references in Line 399-401.

> I appreciate that the effect of nitrate on biofilm formation and motility has been investigated. They are in line with the results reported for *B. pseudomallei* in that nitrate inhibits both biofilm formation and motility and that inactivation of *narL*, the homolog of RSp0980, is no longer responsive to the presence of nitrate. In *B. pseudomallei*, the NarX/NarL two component system

was shown to control of expression adjacent genes encoding enzymes for nitrate-nitrite respiration along with several other functions affecting biofilm formation and secondary metabolite production. As in *B. pseudomallei*, the homologous genes for nitrate respiration are next to RSp0980. It would therefore be important to elucidate its role for nitrate-nitrite respiration, as this appears to be the primary function of RSp0980. In any case, while the authors suggest in the revised version of the manuscript that RSp0980 is possibly involved in nitrate sensing they fail in providing a convincing model in how this would go together with the proposed sensing of 2',3'-cGMP by the same response regulator.

Response: Thanks a lot for your good suggestion, we have added some more discussion in the discussion part "Furthermore, the homologs of RSp0980 in *Burkholderia pseudomallei* and *S. typhimurium* were demonstrated to be involved in nitrate sensing^{48,49}. Bioinformatics analysis showed that Rsp0979 and RSp0980 might also form a two-component system related with nitrate sensing. Besides to be a receptor of 2',3'-cGMP, RSp0980 is possible to receive the signal from the neighboring RSp0979, which needs to be further confirmed (Line 401-Line 405)".

Finally, previous study showed that 2',3'-cNMPs alter bacterial growth by bind to bacterial ribosomes ((Chauhan SS, Marotta NJ, Karls AC, Weinert EE. ACS Cent Sci. 2022 Nov 23;8(11):1518-1526). However, the deletion of RSp0334 had little effect on the growth of bacterial cells in different media (Supplementary Fig. 1C-E) but resulted in significant defects in phenotypes including motility, biofilm formation, cellulase production, and extracellular polysaccharide (EPS) production, and *in trans* expression of RSp0334 restored all of the phenotypes of the RSp0334 deletion mutant to those of the wild-type strain (Fig. 1C-F) (Line107-111), suggesting that 2',3'-cGMP utilized a different mechanism in *R. solanacearum* from that in *E. coli*.

> In fact, the effect on growth reported by Chauhan et al. was not dramatic and is probably comparable with the differences seen between the *Ralstonia* wt and the RSp0334 deletion mutant 0.1 in TTC medium. Moreover, Chauhan et al. showed that a 2-fold increase in the 2',3'-cGMP level does not affect growth. As the effect of deleting RSp0334 in *Ralstonia* increases the 2',3'-cGMP level by less than 2-fold, no effect on growth can be expected and thus an inhibition of translation cannot be excluded.

Response: Good suggestion, we have searched the homologs of ribosomes of *E. coli* in *R. solanacearum*, and found that there is a 30S ribosomal protein S7, 50S ribosomal protein L15, 50S ribosomal protein L11, 50S ribosomal protein L7/L12, 50S ribosomal protein L27, 50S ribosomal protein L28, 50S ribosomal protein L29, 50S ribosomal protein L33, 50S ribosomal protein L34, which share 62.5% to 98.08% with ribosomal proteins of *E. coli*. These proteins might be able to bind to 2',3'-cNMPs. If yes, the ribosomal protein will be another receptor of 2',3'-cGMP in *R. solanacearum*, which is worth to being investigated in the future.

Reviewer #3 (Remarks to the Author):

This is the revised version of a manuscript previously submitted to Nature Communications. The authors have to a great extent satisfactorily answered to my comments. I have the following follow up comments.

The major novel finding is perhaps not the role of 2',3'-cyclic GMP as a second messenger, but the unconventional functionality of RSp0334 in catalyzing 2',3' cyclic GMP to 2',3' cyclic di-GMP. This should be reflected in the abstract.

The phosphorylation site for the NarL homologue RSp0980 on Thr at position 12 is unconventional. Can the authors comment on this in the manuscript.

PdeR of *Salmonella typhimurium* contains a GGDEF motif instead of a GGDQF motif. What is the effect of the E>Q substitution? Does PdeR make both 2',3' cyclic di-GMP and 3',3' cyclic di-GMP? Genetic experiments performed by Ahmad et al, BMC Microbiol, 2017 suggest conventional catalytic cyclase and phosphodiesterase activities by PdeR.

Response: Thanks a lot for your good suggestion. We have emphasized the role of RSp0334 in catalyzing 2',3' cyclic GMP to 2',3' cyclic di-GMP in the abstract as suggested.

Secondly, bioinformatics analysis showed that Rsp0979 and RSp0980 might also form a two-component system related with nitrate sensing. Besides to be a receptor of 2',3'-cGMP, RSp0980 is possible to receive the signal from the neighboring RSp0979 and then will be phosphorylated, which needs to be further confirmed (Line 401-Line 405).

Thirdly, we have described the detailed role of the “Q” residue of the GGDQF domain of RSp0334 in our manuscript from Line 176 to Line 183 “the GGDQF domain of RSp0334 catalyzes 2',3'-cGMP to 2',3'-c-di-GMP, while the GGDEF domain usually catalyzes GTP to bis-3',5'-c-di-GMP, we further investigated the detailed function of the GGDQF domain of Rsp0334. To test whether the Q549 residue is responsible for the distinct functions between the GGDQF domain of RSp0334 and the GGDEF domain of bis-3',5'-c-di-GMP synthase, we then generated a single point mutant, RSp0334 (GGDQF^{Q549E}), in which residue Q549 was substituted by Glu (E) to change the GGDQF domain to a GGDEF domain. Intriguingly, the results showed that RSp0334 (GGDQF^{Q549E}) still exhibited the same activity in catalyzing 2',3'-cGMP but had no activity in converting GTP to bis-3',5'-c-di-GMP (**Supplementary Fig. 6A-B, The response Fig. 10**). These results suggested that E is not the key enzyme active site.”

Fourthly, Ahmad, *et al.* demonstrated the conventional catalytic cyclase and phosphodiesterase activities of STM1703. In this study, we found that PdeR of *Salmonella typhimurium* did not catalyze GTP to bis-3',5'-c-di-GMP, while showed a very slight activity to degrade bis-3',5'-c-di-GMP *in vitro* (**The response Fig. 11**).

The response Fig. 10. Role of the “Q” residue in the GGDQF domain of Rsp0334 (Supplementary Fig. 6A-B)

The response Fig. 11. Degradation of bis-3',5'-c-di-GMP by the PdeR *in vitro*

Reviewer #4 (Remarks to the Author):

The authors have carefully, methodically and convincingly addressed all comments by all 4 Reviewers. The authors have also revised the manuscript according to the suggestions and concerns of all Reviewers, including addition of new data.

Response: Thanks a lot for your valuable and nice comments.

REVIEWERS' COMMENTS

Reviewer #1 (Remarks to the Author):

This second revised manuscript by Li et. al. addresses my concerns by showing intracellular detection of 2'3'-c-di-GMP and its negative correlation with the intracellular concentration of 2'3' cGMP. However, I did not see the methods for how 2'3' c-di-GMP was detected via mass spec. This is very important since this is a novel cyclic di-nucleotide in bacteria. I would also strongly encourage the authors to include some of the raw mass spec trace files in the supplemental data showing detection of 2'3' c-di-GMP as I think there may be pushback on this point in the broader field given that the enzymatic mechanism of synthesis is unclear. Other than those two points, I have no more concerns.

Reviewer #2 (Remarks to the Author):

I am not satisfied how the authors responded to my criticism, namely by deleting false information or adding a few sentences that other explanations are possible but have not been investigated further. Some points were not addressed at all. I still believe that data have been properly analysed or are misinterpreted as exemplified with the identification of RSp0334/2',3'-cGMP/RSp0980 homologues systems in other bacteria:

> The authors claim that the RSp0334/2',3'-cGMP/RSp0980 signaling system also exists in the human pathogen *Salmonella typhimurium*, and may exist in other bacteria. To support this argument the authors present homologs of RSp0334 and RSp0980 in Table S8. However, a closer look at this table shows that of the 11 homologous proteins only those of *Salmonella* and *Ruegeria mobilis* possess the LLARLGGDEF domain while the remaining ones not even have the minimal L542, L545 and D548 motive required for function and thus cannot have a function in 2',3'-cGMP signaling.

It is also not clear how the authors identified homologous proteins. For example, in *P. aeruginosa* the protein with the highest homology to RSp0334 is PA0285, which has a LLSRLGGDEF domain, and not WP_003118137.1 as stated in the table. Notably, several other homologs in PAO1 also have the L542, L545 and D548 motive, some of which are well-characterized enzymes affecting the cellular c-di-GMP level although, according to the definitions of the authors, should be involved in 2',3'-cGMP signaling. Yet, there is no evidence that this would be the case.

Response: Thanks for your good suggestion. We have listed the Table S8 for the convenience of the study of the potential homologs of RSp0334 and Rsp0980 in other bacterial species. We have deleted

this table in this revised version of our manuscript as some of the proteins are not the true homologs of Rsp0334.

The claim that the RSp0334/2',3'-cGMP/RSp0980 signaling system is widespread among bacteria was largely based on the finding that homologs were identified in phylogenetically diverse bacteria. It now turns out that the identified genes are no homologs. While deleting this table is a cheap solution to the criticism raised, it does not consider that this table was a main argument that the signaling system is widespread among bacteria. The argument is now only supported by the finding that four out of 11 strains tested produce 2',3'-cGMP and that *Salmonella typhimurium* ATCC14028 also carries homologs of RSp0334 and RSp0980. Importantly, the other three strains do not contain homologs of RSp0334 (not mentioned in the manuscript!) and thus cannot be used as an argument that the RSp0334/2',3'-cGMP/RSp0980 signaling system is widely present in bacteria.

Another issue that is ignored by the authors is the fact that the homolog of RSp0980 in *S. typhimurium* ATCC14028 is NarL, a very well characterized response regulator of the Nar two-component system, which is critical for nitrate-mediated regulation (Stewart 1982). NarL is not only widely found in prokaryotes and eukarya, but also exists in archaea. Detailed genetic and structural analyses have been performed that showed that the NarL receiver domain operates as an on-off switch to occlude or release the output domain in a nitrate-responsive manner (Katsir et al. 2015). Phosphorylation is the key trigger of NarL to activate or repress the genes encoding nitrate respiratory enzymes (Eldridge et al. 2002). More recently, it was shown that the activity of NarL is also post-transcriptionally controlled via acetylation of two lysin residues affecting the DNA-binding ability of NarL. As pointed out in my previous review RSp0980 is located within a gene cluster encoding enzymes for nitrate respiration. This genetic arrangement is highly conserved in diverse bacteria, strongly suggesting that RSp0980 is a true homolog of the nitrate regulator NarL. Hence, if NarL binds 2',3'-cGMP one would expect that expression of NarL-regulated genes is affected. Given the conservation of the gene cluster encoding NarL its primary function is most likely in nitrate-nitrite respiration. While I have suggested to test this possibility in my previous reviews, this criticism has not been adequately addressed. To add two sentences that "Bioinformatics analysis showed that Rsp0979 and RSp0980 might also form a two-component system related with nitrate sensing. Besides to be a receptor of 2',3'-cGMP, RSp0980 is possible to receive the signal from the neighboring RSp0979, which needs to be further confirmed." does satisfy my concerns. It would certainly be a game changer if unambiguous evidence were presented that 2',3'-cGMP modulates activity of NarL. In this context, it is also noteworthy that NarL has been demonstrated to affect biofilm formation and virulence in different bacteria. I wonder whether the determined RSp0980 binding sites are similar to the consensus sequence of NarL binding? I also wonder whether the mapped NarL regulon contains genes important for nitrate respiration? To this end, it might be necessary to adjust the growth conditions such that nitrate respiration is induced. Finally, the authors state without showing data that 1 mM nitrate and higher concentrations inhibited biofilm formation and motility while lower concentrations had no effect. Previous studies also used nitrate concentrations in the millimolar range and thus this finding indeed substantiates a role of RSp0980 in nitrate sensing.

Another point that still needs clarification is the possibility that 2',3'-cNMPs alters bacterial growth and thus gene expression by binding to bacterial ribosomes, as demonstrated by the Weinert lab and pointed out in my previous reviews. After two revisions, the authors now agree that it is possible that 2',3'-cGMP could also bind to ribosomes in *Ralstonia*. As a response, the authors added the sentence "It was demonstrated that 2',3'-cNMP can bind to bacterial ribosomes and then result in decreased bacterial growth." But neither do the authors provide a convincing argument why this should not be the case in *Ralstonia* nor do they mention that there appears to be an effect on growth, at least under certain the growth conditions, that could support the idea that 2',3'-cNMP binds to ribosomes. This problem is further compounded by the fact NarL is a bona fide nitrate sensor and a convincing model of how this response regulator (which are activated via phosphorylation) at the same time binds 2',3'-cGMP is missing.

Reviewer #3 (Remarks to the Author):

In my opinion, the abstract does not entirely reflect the content of this manuscript. I suggest altering that abstract in the following way:

In-frame deletion of RSp0334, which contains an evolved GGDEF domain with a LLARLGGDQF motif required to catalyze 2',3'-cGMP to (2',5')(3',5')-cyclic diguanosine monophosphate (2',3'-c-di-GMP), altered the abovementioned important phenotypes through increasing the intracellular 2',3'-cGMP levels. Furthermore, we found that 2',3'-cGMP, its receptor and catalytic activity to 2',3'-c-di-GMP exists also in the human pathogen *Salmonella typhimurium*. Together, our work provides new insights into the function of the GGDEF domain of RSp0334 and the regulatory mechanism of 2',3'-cGMP signal in bacteria.

LLARLGGDQF region?, perhaps more accurate: motif

Reviewer #1 (Remarks to the Author):

This second revised manuscript by Li et. al. addresses my concerns by showing intracellular detection of 2'3'-c-di-GMP and its negative correlation with the intracellular concentration of 2'3'-cGMP. However, I did not see the methods for how 2'3'-c-di-GMP was detected via mass spec. This is very important since this is a novel cyclic dinucleotide in bacteria. I would also strongly encourage the authors to include some of the raw mass spec trace files in the supplemental data showing detection of 2'3'-c-di-GMP as I think there may be pushback on this point in the broader field given that the enzymatic mechanism of synthesis is unclear. Other than those two points, I have no more concerns.

Response: Thanks a lot for your good suggestions. We have added more details in the Methods section for the detection of 2',3'-c-di-GMP (Line 514-519). In addition, we have provided the raw mass spec trace files in Source Data file as suggested. The ^{31}P NMR spectra of the produced 2',3'-c-di-GMP (e), ^1H NMR spectra of the produced 2',3'-c-di-GMP (f), the triple-quadrupole mass spectra in an MRM model of the standard 2',3'-c-di-GMP (g) and the produced 2',3'-c-di-GMP (h) were shown in Fig. S3.

Reviewer #2 (Remarks to the Author):

I am not satisfied how the authors responded to my criticism, namely by deleting false information or adding a few sentences that other explanations are possible but have not been investigated further. Some points were not addressed at all. I still believe that data have been properly analysed or are misinterpreted as exemplified with the identification of RSp0334/2',3'-cGMP/RSp0980 homologues systems in other bacteria:

> The authors claim that the RSp0334/2',3'-cGMP/RSp0980 signaling system also exists in the human pathogen *Salmonella typhimurium*, and may exist in other bacteria. To support this argument the authors present homologs of RSp0334 and RSp0980 in Table S8. However, a closer look at this table shows that of the 11 homologous proteins only those of *Salmonella* and *Ruegeria mobilis* possess the LLARLGGDEF domain while the remaining ones not even have the minimal L542, L545 and D548 motive required for function and thus cannot have a function in 2',3'-cGMP signaling.

It is also not clear how the authors identified homologous proteins. For example, in *P. aeruginosa* the protein with the highest homology to RSp0334 is PA0285, which has a LLSRLGGDEF domain, and not WP_003118137.1 as stated in the table. Notably, several other homologs in PAO1 also have the L542, L545 and D548 motive, some of which are well-characterized enzymes affecting the cellular c-di-GMP level although, according to the definitions of the authors, should be involved in 2',3'-cGMP signaling. Yet, there is no evidence that this would be the case.

Response: Thanks for your good suggestion. We have listed the Table S8 for the convenience of the study of the potential homologs of RSp0334 and Rsp0980 in other bacterial species. We have deleted this table in this revised version of our manuscript as some of the proteins are not the true homologs of Rsp0334.

The claim that the RSp0334/2',3'-cGMP/RSp0980 signaling system is widespread among bacteria was largely based on the finding that homologs were identified in phylogenetically diverse bacteria. It now turns out that the identified genes are no homologs. While deleting this table is a cheap solution to the criticism raised, it does not consider that this table was a main argument that the signaling system is widespread among bacteria. The argument is now only supported by the finding that four out of 11 strains tested produce 2',3'-cGMP and that *Salmonella typhimurium* ATCC14028 also carries homologs of RSp0334 and RSp0980. Importantly, the other three strains do not contain homologs of RSp0334 (not mentioned in the manuscript!) and thus cannot be used as an argument that the RSp0334/2',3'-cGMP/RSp0980 signaling system is widely present in bacteria.

Response: Thanks a lot for your good suggestions, we have moved the findings of the homologs of RSp0334 and RSp0980 in *Salmonella typhimurium* ATCC14028 from the result section to the discussion section and modified the relevant descriptions as suggested.

Another issue that is ignored by the authors is the fact that the homolog of RSp0980 in *S. typhimurium* ATCC14028 is NarL, a very well characterized response regulator of the Nar two-component system, which is critical for nitrate-mediated regulation (Stewart 1982). NarL is not only widely found in prokaryotes and eukarya, but also exists in archaea. Detailed genetic and structural analyses have been performed that showed that the NarL receiver domain operates as an on-off switch to occlude or release the output domain in a nitrate-responsive manner (Katsir et al. 2015). Phosphorylation is the key trigger of NarL to activate or repress the genes encoding nitrate respiratory enzymes (Eldridge et al. 2002). More recently, it was shown that the activity of NarL is also post-transcriptionally controlled via acetylation of two lysin residues affecting the DNA-binding ability of NarL. As pointed out in my previous review RSp0980 is located within a gene cluster encoding enzymes for nitrate respiration. This genetic arrangement is highly conserved in diverse bacteria, strongly suggesting that RSp0980 is a true homolog of the nitrate regulator NarL. Hence, if NarL binds 2',3'-cGMP one would expect that expression of NarL-regulated genes is affected. Given the conservation of the gene cluster encoding NarL its primary function is most likely in nitrate-nitrite respiration. While I have suggested to test this possibility in my previous reviews, this criticism has not been adequately addressed. To add two sentences that "Bioinformatics analysis showed that Rsp0979 and RSp0980 might also form a two-component system related with nitrate sensing. Besides to be a receptor of 2',3'-cGMP, RSp0980 is possible to receive the signal from the neighboring RSp0979, which needs to be further confirmed." does satisfy my concerns. It would certainly be a game changer if unambiguous evidence were presented that 2',3'-cGMP modulates activity of NarL. In this context, it is also noteworthy that NarL has been demonstrated to affect biofilm formation and virulence in different bacteria. I wonder whether the determined RSp0980 binding sites are similar to the consensus sequence of NarL binding? I also wonder whether the mapped NarL regulon contains genes important for nitrate respiration? To this end, it might be necessary to adjust the growth conditions such that nitrate respiration is induced. Finally, the authors state without showing data that 1 mM nitrate and higher concentrations inhibited biofilm formation and motility while lower concentrations had no effect. Previous studies also used nitrate concentrations in the millimolar range and thus this finding indeed substantiates a role of RSp0980 in nitrate sensing.

Response: Thanks a lot for your good suggestions. It is possible that Rsp0980 could also sense nitrate in *R. solanacearum*, which is worth exploring in the future. However, this scientific question is not relevant to the title of our manuscript.

Another point that still needs clarification is the possibility that 2',3'-cNMPs alters bacterial growth and thus gene expression by binding to bacterial ribosomes, as demonstrated by the Weinert lab and pointed out in my previous reviews. After two revisions, the authors now agree that it is possible that 2',3'-cGMP could also bind to ribosomes in *Ralstonia*. As a response, the authors added the sentence "It was demonstrated that 2',3'-cNMP can bind to bacterial ribosomes and then result in

decreased bacterial growth.” But neither do the authors provide a convincing argument why this should not be the case in *Ralstonia* nor do they mention that there appears to be an effect on growth, at least under certain the growth conditions, that could support the idea that 2',3'-cNMP binds to ribosomes. This problem is further compounded by the fact NarL is a bona fide nitrate sensor and a convincing model of how this response regulator (which are activated via phosphorylation) at the same time binds 2',3'-cGMP is missing.

Response: Thanks a lot for your good suggestions. It is possible that 2',3'-cGMP could also bind to ribosomes in *R. solanacearum*, which is worth exploring in the future. However, this scientific question is not relevant to the title of our manuscript.

Reviewer #3 (Remarks to the Author):

In my opinion, the abstract does not entirely reflect the content of this manuscript. I suggest altering that abstract in the following way:

In-frame deletion of RSp0334, which contains an evolved GGDEF domain with a LLARLGGDQF motif required to catalyze 2',3'-cGMP to (2',5')(3',5')-cyclic diguanosine monophosphate (2',3'-c-di-GMP), altered the abovementioned important phenotypes through increasing the intracellular 2',3'-cGMP levels. Furthermore, we found that 2',3'-cGMP, its receptor and catalytic activity to 2',3'-c-di-GMP exists also in the human pathogen *Salmonella typhimurium*. Together, our work provides new insights into the function of the GGDEF domain of RSp0334 and the regulatory mechanism of 2',3'-cGMP signal in bacteria.

LLARLGGDQF region?, perhaps more accurate: motif

Response: Thanks a lot for your good suggestions, we have revised these sentences as suggested.